# Applying a Random Time Mapping to Mann modelled turbulence for the generation of intermittent wind fields

Khaled Yassin[1], Arne Helms[2], Daniela Moreno[1], Hassan Kassem[3], Leo Höning[2,3], and Laura J. Lukassen[1]

[1]ForWind, Institute of Physics, Carl von Ossietzky University Oldenburg, Küpkersweg 70, 26129 Oldenburg, Germany
[2]Institute of Physics, University of Oldenburg, Carl von Ossietzky University Oldenburg, Küpkersweg 70, 26129 Oldenburg, Germany
[3]Fraunhofer Institute for Wind Energy Systems – Fraunhofer IWES, Küpkersweg 70, 26129 Oldenburg, Germany

**Correspondence:** Khaled Yassin (khaled.yassin@uni-oldenburg.de), Laura J. Lukassen (laura.lukassen@uni-oldenburg.de)

**Abstract.** A new approach to derive a synthetic wind field model which combines spatial correlations from the Mann model and intermittency is introduced. The term intermittency describes the transition from Gaussian to non-Gaussian velocity increment statistics at small scales, where non-Gaussian velocity increment statistics imply a higher probability for extreme values than a Gaussian distribution. The presented new model is named the Time-mapped Mann model. The intermittency is introduced by applying a special random time-mapping procedure to the regular Mann model. The Time-mapping procedure is based on the so-called Continuous-time random walk model. As will be shown, the new Time-mapped Mann field reflects spatial correlations from the Mann model in the plane perpendicular to flow direction and temporal intermittency. In the first wind turbine study, the new Time-mapped Mann field and a regular Mann field are used as inflow to a wind turbine in a Blade Element Momentum simulation. It is shown that the wind field intermittency carries over to loads of the wind turbine, and, thus, shows the importance of carefully modeling synthetic wind fields.

## 1 Introduction

Wind energy plays a leading role in the energy transition process to renewable energy these days. In 2020, the world witnessed new wind energy installations by 93 GW making the global installed capacity of wind energy 743 GW according to the Global Wind Energy Council (GWEC (2021)). The growing demand for wind energy resulted in growing turbine rotor diameters. This leads to an increase in loads on the different turbine components like blades and tower. Numerical simulations are now crucial to predict loads and performance parameters in the early design stages. In this context, wind fields should be accurately simulated to be able to anticipate extreme load cases and fatigue loads on different parts of the wind turbine to reach the optimum design without compromising the structural limits of these components. However, the physics of the turbulent wind fields are not yet completely understood and many models were proposed to simulate these fields as described in the following.

The IEC 61400-1 standard for wind turbines (Han (2007)) recommends using Gaussian based atmospheric turbulence models, namely Kaimal (Kaimal et al. (1972)) and Mann (Mann (1994, 1998)) models to simulate turbulent wind fields. These two turbulent wind field models imply the assumption that the velocity increments of the wind fields behave in a Gaussian manner. However, it is well documented that the velocity increments of real atmospheric wind fields show non-Gaussian behavior, which is widely called intermittency. Through the heavy tails of the distribution, this implies a larger probability for extreme values than in a Gaussian distribution. Boettcher et al. (2003) showed that wind field measurements from near the German North Sea coastline have a non-Gaussian distribution of two-point statistics of velocity increments for certain mean wind velocity intervals. Also, Vindel et al. (2008) have shown the intermittency of velocity increment statistics in data of the atmospheric boundary layer velocity measured by Cuxart et al. (2000). More recent researches like the work of Liu et al. (2010) and Mücke et al. (2011) reached the same conclusion. This means that extreme wind velocity increments happen more often than what is predicted using the turbulence models recommended by the IEC standard. Also, Morales et al. (2012) presented a statistical scheme to assess turbulent wind fields that uses both, lower and higher-order one-point and two-point statistics. They have also compared synthetic turbulent time series using the Kaimal model with atmospheric turbulent wind field measurements from the research platform FINO-1. They have shown that the synthetic fields generated with the Kaimal model failed to grasp the intermittent behavior of the wind field.

The questions that arise at this point are, how a synthetic wind field can be modeled as realistically as possible and how this is reflected in the turbine loads. In this direction, Kleinhans (2008) proposed a new approach to generate an intermittent wind field using the Continuous Time Random Walk (CTRW) theory. This approach is based on stochastic differential equations and function that maps from an intrinsic time to real-time to generate such an intermittent field whereas the spatial correlations are not as realistic as in the Mann field. A brief description of the CTRW model will be introduced in Sec. 2.3. In addition to the CTRW model, there also exist other models which can be applied to generate synthetic wind fields with intermittent features, e.g. by Friedrich et al. (2021) based on multi-point statistics which are not further discussed here.

Gontier et al. (2007) compared the effect of the intermittent CTRW wind field and other standard turbulence models (Kaimal and von Kármán (1948)) on the fatigue loads of the D8 small wind turbine at different wind speeds. The authors have found that the simulated wind turbine loads in the case of using the CTRW wind model show heavy-tailed second-order statistics, while the loads simulated by using the other two models do not show such a behavior. However, the authors could not give definitive conclusions regarding the peak loads from their simulations. Later, Mücke et al. (2011) compared the statistics of wind fields generated by the Kaimal model, intermittent wind fields generated using CTRW, and the atmospheric wind data measured within the GROWIAN project (Körber et al. (1988)). The three cases were then used in the NREL FAST (V6.01) (Jonkman and Jonkman (2005)) Blade Element Momentum (BEM) simulator to model the resulting rotor torque of the 1.5MW WindPACT (Malcolm and Hansen (2006)) turbine. Also, the authors analyzed the stress cycles for the rotor torque time series to study the effects of the three wind fields on fatigue loads. They have concluded that the statistics of the numerical results using the Kaimal wind field qualitatively differ from the results obtained from measurements. On the other hand, they have concluded that the synthetic wind field generated by the CTRW managed to reproduce the intermittency at small time scales. Also, the CTRW field managed to qualitatively provide intermittent rotor torque statistics like the resulting torque from mea-

surements.

Gong and Chen (2014) numerically investigated the short- and long-term extreme responses of the onshore NREL 5MW turbine for a Gaussian wind field using a Kaimal wind spectrum and a second non-Gaussian wind field generated by a translation process theory introduced by Gioffre et al. (2000). They showed that the non-Gaussian wind field resulted in larger extreme loads of the blade root edge-wise and tower fore-aft moments while the responses of a stand-still wind turbine case were less sensitive to the non-Gaussianity of the wind field. Schottler et al. (2017) applied an intermittent wind field on a 0.58m diameter model wind turbine in a wind tunnel to measure the effect of such a wind field on the thrust, power, and torque on the rotor. The authors found out that the turbine did not smooth out the intermittency and they also concluded that assuming non-Gaussian wind fields may lead to significant differences in loads on the turbine.

Schwarz et al. (2018, 2019) isolated and investigated the effect of intermittent wind velocity on equivalent fatigue loads of the NREL 5MW wind turbine at different wind speeds using BEM simulations. To isolate the effects of intermittency, two different velocity fields were generated utilizing the CTRW model: a Gaussian and a non-Gaussian wind field. Both of them have the same features except for the intermittency that was introduced in the non-Gaussian field utilizing the time mapping, which was simply omitted in the Gaussian case. For the two wind cases, three different spatial correlations were investigated: fully correlated fields, delta-correlated fields, and 3×3 sub-divided fully correlated fields. After analyzing the results of the different cases, the authors noticed that the highest effect of intermittency of the wind fields on the loads can be observed in the fully correlated case with 5% to 10% increase in fatigue loads while it completely disappears in the delta-correlated field. For the 3×3 sub-divided fully correlated field, intermittent loads were ranged between the other two cases. The authors concluded that intermittency in the wind field had a significant effect on loads.

Berg et al. (2016) also compared Gaussian and non-Gaussian wind fields using Large Eddy Simulations (LES) in the HAWC2 Computational Fluid Dynamics (CFD) software (Larsen and Hansen (2007)). They studied the effect of the intermittency on the different blade, tower, and shaft loads and deflections of the NREL 5MW wind turbine. In contrast with the aforementioned studies, they concluded that intermittency has no significant effect on the studied parameters. This shows that even though many studies reveal the influence of intermittency, the effect needs further investigation. Recently, Bangga and Lutz (2021) simulated the DANAERO 2MW wind turbine under turbulent inflow using CFD and BEM simulations. In their work, they have compared tip deflections, flap-wise and edge-wise loads on the blade root, and the damage equivalent load (DEL) of the DANAERO wind turbine affected by a synthetic turbulent Mann field. The results of the simulated loads and deflections in their research have shown good agreement with measured data from the turbine operating under the same conditions. Specifically, the authors compared the flapwise, edgewise, and torsional deflections' spectra of the turbine under laminar and turbulent inflows. The presented analysis of the spectra is very detailed. However, the spectra of the deflections are not enough to study all aspects of the effects of turbulence.

In the present paper, a novel method to numerically generate a synthetic, intermittent turbulent wind field is introduced. Within this new method, the time-mapping technique introduced by Kleinhans (2008) is applied on a turbulent wind field generated by the Mann model. A detailed comparison between our new wind field using this time-mapping technique referred to as the Time-mapped Mann field in the following, and a standard Mann-modeled wind field is carried out. As will be shown, the ad-

vantage of the new Time-mapped Mann field is that it combines spatial correlations from the Mann wind field and intermittent behavior. Also, a first insightful comparison between selected mechanical loads on a 1.5MW wind turbine resulting from both, the Time-mapped Mann field and the standard Mann field is illustrated. The analysis of these turbine loads reveals the effect of intermittency on wind turbine loads.

95

## 2 Scientific background on synthetic wind fields

Before illustrating the derivation of the new Time-mapped Mann field, general wind field statistics that will be used to analyze the generated wind fields are introduced in Sec. 2.1. After that, in 2.2 and 2.3 the Mann and CTRW turbulent wind field models are introduced. The main purpose of this section is to provide the theoretical background for the derivation of the new model

and its analysis. This chapter is also intended to be self-contained and provide all the necessary equations for the introduction of the proposed new Time-mapped Mann model.

### 2.1 Statistics of wind fields

For any turbulent wind field, the wind velocity time series $U_i$ can be expressed as a function of mean velocity $\langle U_i \rangle$ and velocity fluctuation $u_i$, with $i = 1, 2, 3$ in three-dimensional space:

$$U_i = \langle U_i \rangle + u_i, \tag{1}$$

where the angular brackets $\langle . \rangle$ denote an ensemble average. Throughout this paper, the ensemble average is partly replaced by spatial or temporal averages for practical purposes which will be indicated in the respective cases below. In our case, we assume only a mean velocity in $x_1$-direction, i.e. $\langle \boldsymbol{U} \rangle = (\langle U_1 \rangle, 0, 0)$ which is the inflow direction in the following. From this decomposition, the turbulence intensity (TI) can be calculated as follows:

$$TI_i = \frac{\sigma_i}{\langle U_1 \rangle} = \frac{\sqrt{\langle u_i^2 \rangle}}{\langle U_1 \rangle}, \tag{2}$$

where $i$ indicates the direction in which the turbulence intensity is calculated with respect to the mean velocity in the inflow component $\langle U_1 \rangle$, and $\sigma_i$ as the variance of $u_i$. The turbulence intensity is a one-point statistics in space and time of wind fields. However, for information on the spatial structures, one-point statistics is not sufficient and two-point statistics should be used for more information about the wind fields. As an example, the co-variance tensor contains two-point statistics:

$$R_{ij}(\mathbf{r}, \boldsymbol{x}, t) = \langle u_i(\boldsymbol{x}, t) u_j(\boldsymbol{x} + \mathbf{r}, t) \rangle \tag{3}$$

where $R_{ij}$ is the two-point correlation which is independent of the position $\boldsymbol{x}$ in case of homogeneous turbulence and $\boldsymbol{r}$ is the displacement vector between the two points. Since Eq. 3 is a theoretical equation, there are no restrictions on selecting the two points for the two-point statistics. However, in practice, it is limited to the size of the data set of the wind fields. The spectral

velocity tensor for homogeneous turbulence resulting from a Fourier transform of Eq. (3) gives (Pope (2001)):

$$\Phi_{ij}(\boldsymbol{\kappa},t) = \frac{1}{(2\pi)^3} \int\limits_{-\infty}^{\infty} \int\limits_{-\infty}^{\infty} \int\limits_{-\infty}^{\infty} R_{ij}(\mathbf{r},t)\exp(-i\boldsymbol{\kappa}\cdot\mathbf{r})d\mathbf{r}. \tag{4}$$

In this equation, $\boldsymbol{\kappa} = (\kappa_1, \kappa_2, \kappa_3)$ represents a three-dimensional wavenumber vector for the three directions. The one-dimensional spectrum is following from Eq. (4) by integration:

$$F_i(\kappa_1,t) = \int\limits_{-\infty}^{\infty}\int\limits_{-\infty}^{\infty} \Phi_{ii}(\boldsymbol{\kappa},t)d\kappa_2 d\kappa_3 = \frac{1}{2\pi}\int\limits_{-\infty}^{\infty} R_{ii}(r_1,0,0,t)\exp(-i\kappa_1 r_1)dr_1 \tag{5}$$

where the $ii$-index refers to the respective diagonal element of the tensor. The one-dimensional spectra with respect to $\kappa_2$ and $\kappa_3$ are computed accordingly. The spectral coherence for the wavenumber $\kappa_1$ with respect to two separate points in the $x_2 - x_3$ plane reads (Mann (1994)):

$$\mathrm{coh}_{ij}(\kappa_1, \Delta x_2, \Delta x_3, t) \equiv \frac{|\chi_{ij}(\kappa_1, \Delta x_2, \Delta x_3, t)|^2}{F_i(\kappa_1,t)F_j(\kappa_1,t)} \tag{6}$$

with $F_i(\kappa_1,t)$ from equation (5) and the cross spectra $\chi_{ij}$ as defined below:

$$\chi_{ij}(\kappa_1, \Delta x_2, \Delta x_3, t) = \frac{1}{2\pi}\int\limits_{-\infty}^{\infty} R_{ij}(r_1, \Delta x_2, \Delta x_3)\exp(-i\kappa_1 r_1)dr_1 \tag{7}$$

$$= \int\limits_{-\infty}^{\infty}\int\limits_{-\infty}^{\infty} \Phi_{ij}(\boldsymbol{\kappa},t)\exp(i(\kappa_2\Delta x_2 + \kappa_3\Delta x_3))d\kappa_2 d\kappa_3. \tag{8}$$

where $\Delta\boldsymbol{x}$ indicates the spatial step in three-dimensional space with its components $\Delta x_1$, $\Delta x_2$, and $\Delta x_3$ in $x_1$-, $x_2$-, and $x_3$-directions, respectively. Another example of two-point statistics is the increment statistics in space described by:

$$\tilde{v}_i(\boldsymbol{x}, \Delta\boldsymbol{x}, t) = u_i(\boldsymbol{x} + \Delta\boldsymbol{x}, t) - u_i(\boldsymbol{x}, t). \tag{9}$$

Instead of spatial increments, one can also consider temporal increments as

$$v_i(\boldsymbol{x}, t, \tau) = u_i(\boldsymbol{x}, t+\tau) - u_i(\boldsymbol{x}, t) \tag{10}$$

where $\tau$ indicates the time lag. The probability density functions (PDFs) of the introduced temporal increments $v_i$ have been investigated (Vindel et al. (2008), Liu et al. (2010), Mücke et al. (2011), Böttcher et al. (2007), Muzy et al. (2010)) as part of a more detailed characterization of wind turbulence beyond the standard guidelines. The Gaussian distribution is completely described by the mean and the variance. Further statistical information is required for characterizing a non-Gaussian distribution. The so-called kurtosis of a distribution allows quantification of its deviation from a Gaussian distribution. For the temporal increments $v_i$, the kurtosis is defined as:

$$\mathrm{Kurt}(v_i)(\boldsymbol{x}, t, \tau) = \frac{\langle v_i(\boldsymbol{x}, t, \tau)^4 \rangle}{\langle v_i(\boldsymbol{x}, t, \tau)^2 \rangle^2}. \tag{11}$$

For a Gaussian distribution, the kurtosis equals 3. Higher values indicate a heavy-tailed distribution in which extreme velocity increment values have a higher probability than predicted by a Gaussian distribution. We assume statistically stationary turbulence and, hence, omit the time $t$ in the equations in the following.

Fig. 1 shows characteristics of temporal velocity increment statistics of atmospheric wind speed measurements. The data were collected at the site of a Nordex wind turbine with a hub height of 125m, located in Northern Germany. Time series of 10 minutes in length were considered for the analysis. These time series correspond to the horizontal magnitude of the wind speed measured at the hub height with a sampling rate of 50Hz; the vertical component ($U_3$) was neglected here. In total, 494 time series were provided. Similar to Eq. 1, the measured wind speed can be decomposed as $U_{meas} = \langle U_{meas} \rangle + u_{meas}$, where $\langle U_{meas} \rangle$ is calculated over the 10-minutes period. The turbulence intensity for the measured data is calculated as $TI_{meas} = \sqrt{\langle u_{meas}^2 \rangle}/\langle U_{meas} \rangle$. All the mean values of the wind direction, calculated over each of the 10 minutes periods, are distributed within a range of 165°. The selection and preparation of the data set were performed by Nordex according to internal objectives from the analysis. Here, we used the wind data set for illustration purposes rather than for a rigorous investigation of its characterization. Nevertheless, for our study, we investigate the time series of the wind velocity with similar first and second-order moments. Accordingly, we selected a subset of time series whose values of mean wind speed and turbulence intensity are contained within a specific range. The range is defined as $9 \pm 1\mathrm{m/s}$ for the mean wind speed and $0.15 \pm 0.02$ for the turbulence intensity. Further, we also constrain the data in terms of their wind direction. For that, we consider the mean direction calculated over the 10-minute period. Then, we define a range of $\pm 20°$ from the main wind direction at the specific site. After the selection process, 20 out of the 497 available time series are analyzed.

The increments for the atmospheric measured data $v_{meas} = u_{meas}(\mathrm{x}, \, t + \tau) - u_{meas}(\mathrm{x}, t)$ were calculated similar to Eq. (10). Then, the statistics of $v_{meas}$ are computed for the whole subset of 20 conditioned 10-minutes time series. Fig. 1(a) presents the PDFs of $v_{meas}$ for different time lags $\tau$ from 1sec to 60sec. For clarity of presentation, individual distributions are depicted with different markers and shifted vertically. For comparison, all the PDFs are normalized to a standard deviation equal to 1 and the corresponding Gaussian distributions are shown by a solid line.

As can be observed, the PDFs of $v_{meas}$ for time scales $\tau$ equal to 1sec, 5sec and 10sec deviate from the respective Gaussian distributions. Specifically, the distributions of $v_{meas}$ show heavy tails which represent the high probability of extreme events, which is higher than for the Gaussian PDFs. This effect, which is called intermittency, is a well-known feature of atmospheric wind as Vindel et al. (2008), Liu et al. (2010), Mücke et al. (2011), Böttcher et al. (2007) and many others have found. Similar to the results from the investigations mentioned earlier, Fig. 1(a) shows that the temporal increments $v_{meas}$ are intermittent over a broad range of time lags $\tau$. The evolution of the intermittency with $\tau$ can be quantified from the kurtosis. Fig. 1(b) shows the calculated kurtosis of the PDFs according to Eq. (11) of the atmospheric increments $v_{meas}$ as a function of $\tau$. The values of Kurt($v_{meas}$) higher than 3 reveal the intermittent behavior of the atmospheric data over a wide range of $\tau$. For the analyzed data, intermittent features are distinguishable up to time scales of $\tau \approx 11$sec when Kurt($v_{meas}$) converges to the Gaussian value of 3.

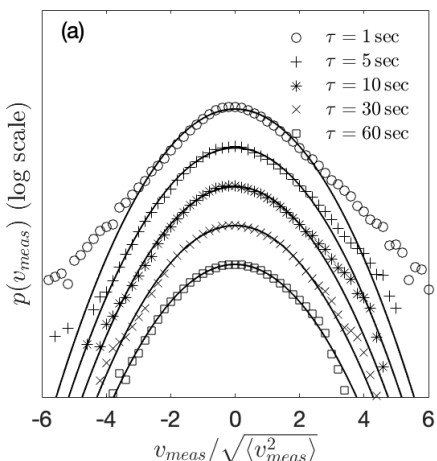 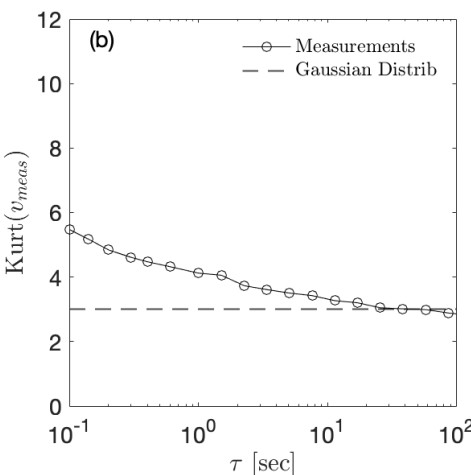

**Figure 1.** Statistical description of turbulent wind measurements from a Nordex wind turbine located in Northern Germany. (a) Normalized PDFs of velocity increments $v_{meas}$ for different values of $\tau$. The distributions for different values of $\tau$, increasing from top to bottom, are depicted by different markers. In addition, solid lines correspond to Gaussian distributions with identical mean and standard deviation. In principle, the curves would all lie on top of each other but here, the curves are shifted vertically for better visualization. (b) Kurtosis of velocity increments $v_{meas}$ as a function of $\tau$. The angular brackets denote moving averages within the individual 600 s-time series and then over all data sets.

## 2.2 Mann Model

The basis of the Mann model (Mann (1994, 1998)) is a properly modeled spectral tensor according to Eq. (4). In this model, the turbulent velocity field is assumed to be incompressible and the velocity fluctuations are assumed to be homogeneous in space. The resulting velocity field depends on only three parameters, namely a length-scale $L$, a non-dimensional parameter Γ which is related to the eddy lifetime and, hence, to the shear gradient, and a parameter $c_K \epsilon^{2/3}$ with $c_K$ as a constant and $\epsilon$ as the turbulent dissipation rate per unit mass. Within these assumptions, the second-order statistics such as variances and cross-spectra of real wind can be met. In case of isotropic turbulence (where assuming no shear) the spectral tensor takes the form (Pope (2001)):

$$\Phi_{ij}^{iso}(\boldsymbol{\kappa}) = \frac{E(\kappa)}{4\pi\kappa^4}\left(\delta_{ij}\kappa^2 - \kappa_i\kappa_j\right) \tag{12}$$

with $\kappa$ as the magnitude of $\boldsymbol{\kappa}$ and $E(\kappa)$ as the energy spectrum. $E(\kappa)$ can be expressed with $c_K\epsilon^{2/3}$ and $L$ that leads to (von Kármán (1948), Pope (2001)):

$$\Phi_{ij}^{iso}(\boldsymbol{\kappa}) = \frac{c_K\epsilon^{2/3}L^{17/3}}{4\pi}\frac{\delta_{ij}\kappa^2 - \kappa_i\kappa_j}{[1 + (L\kappa)^2]^{17/6}}. \tag{13}$$

The corresponding one-dimensional spectra result in (Mann (1994)):

$$
F_i(\kappa_1) = \begin{cases} \frac{9}{55} c_K \varepsilon^{2/3} \frac{1}{(L^{-2}+\kappa_1^2)^{5/6}} & \text{for } i = 1 \\ \frac{3}{110} c_K \varepsilon^{2/3} \frac{3L^{-2}+8\kappa_1^2}{(L^{-2}+\kappa_1^2)^{11/6}} & \text{for } i \in \{2,3\}, \end{cases}
$$

(14)

accordingly, the variance becomes:

$$
\sigma_{iso}^2 = \sigma_1^2 = \sigma_2^2 = \sigma_3^2 = \frac{9}{55} \frac{\sqrt{\pi}\Gamma(\frac{1}{3})}{\Gamma(\frac{5}{6})} c_K \epsilon^{2/3} L^{2/3} \approx 0.688 c_K \epsilon^{2/3} L^{2/3}
$$

(15)

where in this equation the $\Gamma$ denotes the mathematical Gamma function. The value of the $c_K \epsilon^{2/3}$ parameter can be found by fitting the model to specific site data assuming isotropic turbulence and an infinitely large domain. Another possible way to avoid such assumptions was suggested by Larsen and Hansen (2007) who suggest choosing the value of $c_K \epsilon^{2/3}$ arbitrarily

and then re-scaling the velocity field with a scaling factor. In this work, the fields used in the simulations were not re-scaled. However, the value of $c_K \epsilon^{2/3}$ was taken according to the Engineering Sciences Data Unit (ESDU (1982)) spectral model to achieve $TI_1 \approx 10\%$ as will be explained later in detail.

The Mann model provides a spatially correlated synthetic turbulent wind field where spatial homogeneity is one of the basic assumptions in deriving this model. However, since this model is based on the spectra but not on increment statistics of the

atmospheric wind fields, it fails to display intermittency which should have effects on several turbine loads as shown in different former researches, cf. Sec. 1.

### 2.3  Continuous time random walk (CTRW) model

Kleinhans (2008) introduced the idea of generating a synthetic velocity time series $\mathbf{u}(s)$, on an intrinsic time scale $s$, based on a coupled Ornstein-Uhlenbeck process (Uhlenbeck and Ornstein (1930)) for $\mathbf{u}(s)$ and a reference wind speed $\mathbf{u_r}(s)$. Instead

of a mean velocity, this reference wind speed $\mathbf{u_r}(s)$ is used in the modeling process which influences the whole velocity field (Kleinhans (2008)).

After generating the velocity time series $\mathbf{u}(s)$ for all grid points in a plane perpendicular to the flow velocity, a stochastic process is applied to map the field from the intrinsic time $s$ to the physical time domain $t$. This mapping process is the crucial feature of the model for generating intermittent characteristics in the wind field. The mapping process is defined as (Fogedby

(1994)):

$$
\frac{\mathrm{d}t(s)}{\mathrm{d}s} = \tau_\alpha(s)
$$

(16)

where $\tau_\alpha$ follows an $\alpha$-stable Lévy distribution with characteristic exponent $\alpha$ (Kleinhans and Friedrich (2007)), for general parameterization cf. Metzler and Klafter (2000):

$$
p(\tau_\alpha) = \frac{1}{\pi} \mathrm{Re} \left\{ \int_0^\infty \mathrm{d}z \exp\left[ -\mathrm{i}z\tau_\alpha - z^\alpha \exp\left( -\mathrm{i}\frac{\pi\alpha}{2} \right) \right] \right\}.
$$

(17)

Kleinhans (2008) generated the random variable $\tau_\alpha$ according to the implementation introduced by Weron (2001):

$$\tau_\alpha = \frac{\sin(\alpha(V + \frac{\pi}{2}))}{\cos(V)^{\frac{1}{\alpha}}} \left( \frac{\cos(V - \alpha(V + \frac{\pi}{2}))}{W} \right)^{\frac{1-\alpha}{\alpha}} \qquad (18)$$

where $V$ is a uniformly distributed random variable that takes a value in the range between $]-\frac{\pi}{2}, \frac{\pi}{2}[$ and $W$ is an exponentially distributed variable with a mean value = 1. The Lévy distribution is truncated for $0 < \tau_\alpha < c$ if $0 < \alpha < 1$, where $c$ is a cutoff. The $\alpha$-stable Lévy process with the parametrization as in Eq.17 ensures that the $\tau_\alpha$ does not take negative values. Kleinhans
(2008) showed that when continuous stable Lévy processes are used for the transformation from the intrinsic time $s$ to the physical time $t$, the process is dominated by "waiting" regions. In this context, these regions are periods during which the wind speed is constant, a feature that is not observed in the atmospheric wind. In order to limit those periods to a realistic length, the Lévy distribution is truncated at the cutoff $c$. When $\alpha = 1$, $p(\tau_\alpha)$ is not a Lévy distribution anymore. It becomes a $\delta$-correlated distribution with $\tau_1 = 1$. In that case, the mapping process from $s$ to $t$ is linear which makes $\mathbf{u}(s) = \mathbf{u}(t)$ and no intermittent
behavior is introduced to the field.

As mentioned before, the main advantage of the CTRW model is that it manages to generate an intermittent wind field. However, the original CTRW model proposed by Kleinhans (2008) assumes that the spatial correlations of the velocity fluctuations between different points of the grid decay exponentially with the distance between them. This assumption must be understood as a simplification of atmospheric turbulence. A modified version of the original CTRW model was implemented by Schwarz
et al. (2019) as introduced in Sec. 1. They generated individual CTRW time series and arranged them to generate three different spatial correlations in the wind field. Firstly, they set up fully correlated wind fields by repeating the same velocity time series in the whole $x_2$-$x_3$ plane (perpendicular to the inflow direction). For the second arrangement, they generated delta-correlated wind fields by setting up an independent Ornstein-Uhlenbeck process in each point of the grid in the $x_2$-$x_3$ plane. Finally, they considered an intermediate version of a 3×3 sub-divided fully correlated wind field. In this third scenario, the rotor plane was
divided in a 3×3 grid in which each cell corresponds to a fully correlated area, but the nine areas are uncorrelated between them. They found different effects of the wind fields on the turbine loads depending on the resolution (i.e. fully correlated, delta-correlated, or partially correlated). However, no conclusive recommendation was provided on which resolution is most realistic.

## 3 The new Time-mapped Mann model

The new proposed Time-mapped Mann model aims at maintaining the spatial correlation provided by the Mann model explained in Sec. 2.2 while introducing the intermittency employing the time-mapping process of the CTRW model explained in Sec. 2.3. Accordingly, in our new Time-mapped Mann model, the Ornstein-Uhlenbeck process used to generate the wind field in the original CTRW model is replaced with a Gaussian wind field using Mann's model. The longitudinal spatial dimension in the Mann model is converted into the intrinsic time according to the Taylor hypothesis (Taylor (1938)). After that, the time
mapping from intrinsic time $s$ to physical time $t$ is applied to each step $n$ using the discretized form of Eq. (16) (Kleinhans

(2008)):

$$t(s_{n+1}) = t(s_n) + C_{\alpha,c,\Delta s_t}\tau_\alpha(s_n) \tag{19}$$

where $C_{\alpha,c,\Delta s_t}$ is a normalization factor that has the dimension of a time and depends on the Lévy exponent $\alpha$, the cutoff $c$ and the discretization parameter $\Delta s_t = s_{n+1} - s_n$. The intrinsic time step $\Delta s_t$ is equidistant while the physical time step $\Delta t_n$ is non-equidistant. The factor $C_{\alpha,c,\Delta s_t}$ is introduced so that the mean slope of $t(s)$ is one, which means that the development of the physical time and the intrinsic time occurs statistically at the same speed.

As shown in Fig. 2, the Time-mapped Mann model generates a turbulent wind field which is spatially correlated in the two transversal directions ($x_2$ and $x_3$) since it is not changed from the original Mann model besides an interpolation procedure explained below. On the other hand, the conversion between the intrinsic and physical time scales using the random $\tau_\alpha$ leads to intermittent behavior in the longitudinal direction $x_1$ (or in other words in the temporal behavior since the longitudinal direction $x_1$ is converted to time with Taylor's hypothesis), as will be shown in Sec. 4. Accordingly, it presents a compromise between Gaussian, non-intermittent fields generated using the Mann model and the intermittent CTRW model.

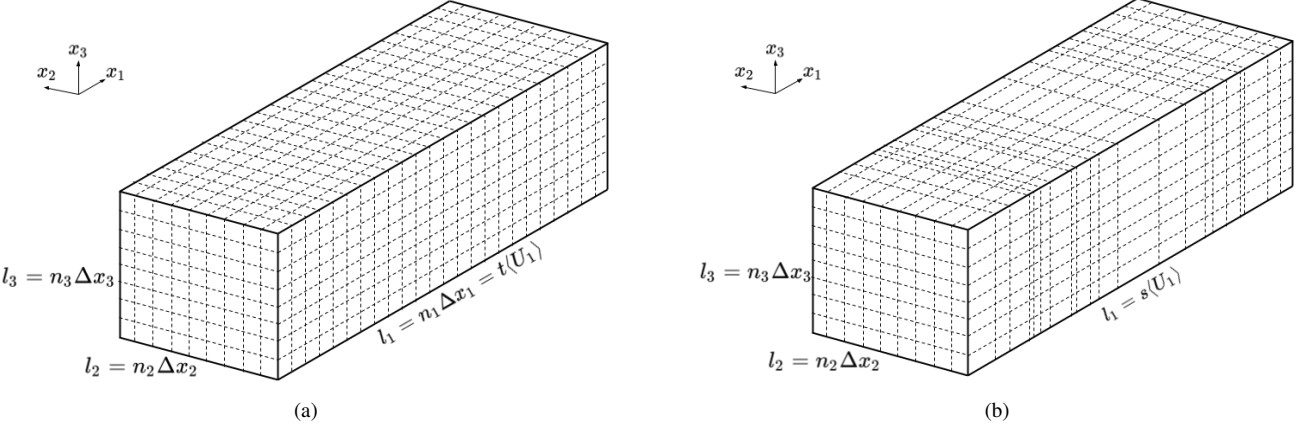

**Figure 2.** Schematic illustration of turbulence boxes generated by the (a) Mann model with equidistant steps and (b) Time-mapped Mann model after applying the time mapping process resulting in non-equidistant steps.

The effect of the generated intermittent wind field on loads of a wind turbine is analyzed using a numerical BEM simulation (described in Sec. 4.2). This numerical model requires that the size of the time step $\Delta t$ of the incoming wind field is equidistant during the simulation. However, as schematically shown in Fig. 2, the time mapping process described by Eq. (19) generates non-equidistant time steps $\Delta t_n$, proportional to $\tau_\alpha$ calculated with Eq. (18). Therefore, the wind speed $U_i$ has to be linearly interpolated between the discrete points obtained with the random numbers $\tau_\alpha$ to the required discrete points distanced by a fixed time increment $\Delta t_{BEM}$. An example of this interpolation is graphically illustrated in Fig. 3. The dashed vertical red lines correspond to the discrete times from the mapping process separated by $\Delta t_n \propto \tau_\alpha$. The red dots show the process $U_i$ at those discrete times. Similarly, the continuous vertical black lines depict the times equally distanced by $\Delta t = \Delta t_{BEM}$. Then, the

linear interpolation takes place between the red dots to obtain the values of $U_i$ at the times depicted by the black lines. The resulting equally distanced discrete points are shown by the black squares. For clarification purposes, the schematic illustrations shown in Fig. 2 and Fig. 3 do not correspond to the same time-mapping process but should just demonstrate the mathematical process.

To generate a Mann box, the HAWC2 Mann turbulence generator tool (Larsen and Hansen (2007)) is used. After that, an in-house Matlab (MATLAB (2019)) code is used to create the Lévy distribution for $\tau_\alpha$ to apply the Time-mapping and interpolation processes.

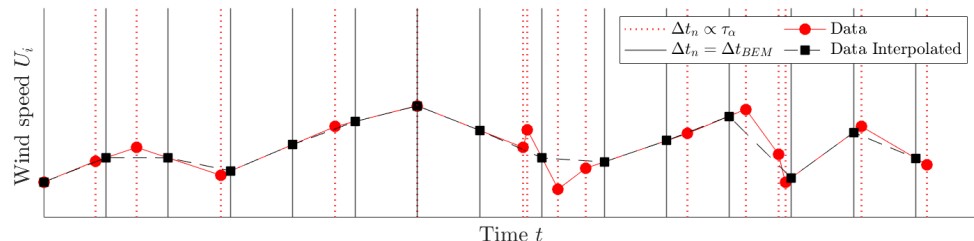

**Figure 3.** Schematic illustration of the interpolation from non-equally distributed time increments $\Delta t_n \propto \tau_\alpha$ to equally distributed time increments $\Delta t_{BEM}$ for the BEM simulations.

## 4 Results and Discussion

In this section, the analysis of the new Time-mapped Mann wind field and exemplary resulting loads from this time mapping 275 will be shown. To clarify the impact of intermittency on the wind fields and loads in this work, all cases will be compared to the original Mann wind field generated with the same parameters to provide a fair comparison between the two models.

### 4.1 Characteristics of the Time-mapped Mann model

This section provides a thorough comparison of statistical quantities such as the spectral tensors, the coherence, and the velocity increment distribution of both fields, the regular Mann and the new Time-mapped Mann wind field. The dimensions of the grid 280 for the generation of the wind fields are selected according to Table 1, such that they can be used for the BEM simulations in Sec. 4.2. In this case, the defined grid has $32 \times 32$ nodes with $\Delta x_{2,m} = \Delta x_{3,m} = 2.6m$ covering an $80.6m \times 80.6m$ area. The original Mann field is generated using the Mann model at mean wind speed $\langle U_1 \rangle = 20m/s$, turbulence intensity in the flow direction, which is $x_1$-direction in this case, $TI_1 = 10\%$, $\Gamma = 0$, and $c_K \epsilon^{2/3} = 0.62 \ m^{4/3}/s^2$ according to the parameters of the Mann model as introduced in the ESDU spectral model. The value of $TI_1 = 10\%$ was used in this work to keep the same 285 standard deviation value as used for the class B turbine (average class) in the IEC 61400-1 standard (Han (2007)) at $20m/s$ mean wind speed. Corresponding to $\Gamma = 0$, we assume homogeneous turbulence, and wind shear is not taken into consideration in this work. Also, based on this assumption, the velocity components can be averaged over each slice to calculate the wind

field statistics. According to Kelly (2018), the $L$ parameter can be calculated by:

$$L = z \frac{TI_1}{a} \tag{20}$$

where $z$ is the height of the calculation point (which is the turbine hub height in this work) and $a$ is a shear exponent. Eq. 20 is used to calculate the length scale for multi-megawatt wind turbines (Hannesdóttir et al. (2019)). However, this equation is also used in the present work for a 1.5MW wind turbine in order to enable a generalization for larger wind turbines. Even though we assume no shear in this work, the shear exponent value $a$ is only used to calculate the value of $L$ with no effect on the wind velocities. According to the hub height of the studied 1.5MW turbine with $z = 84m$, and for neutral conditions with a shear exponent of $a = 1/7$ (Schlichting and Gersten (2016)), Eq. (20) results in $L = 58.8m$.

**Table 1.** Mann box and time-mapping parameters for the generated wind fields.

| $n_{x_1}$ | $n_{x_2} = n_{x_3}$ | $\Delta x_{1,m}$ | $\Delta x_{2,m} = \Delta x_{3,m}$ | $\Gamma$ | $L$ | $c_K \epsilon^{2/3}$ | $TI_1$ | $\langle U_1 \rangle$ | $\alpha$ |
|---|---|---|---|---|---|---|---|---|---|
| $[-]$ | $[-]$ | $[m]$ | $[m]$ | $[-]$ | $[m]$ | $[m^{4/3}/s^2]$ | $[\%]$ | $[m/s]$ | $[-]$ |
| $1.31072 \times 10^5$ | 32 | 2 | 2.6 | 0 | 58.8 | 0.62 | 10 | 20 | 0.6 |

For the Time-mapped Mann wind field, the three components of the velocity fluctuations of each transverse plane ($x_2$-$x_3$ in this case) are shifted according to the resulting $t(s)$. When applying the time mapping process to the wind field, the same stochastic process ($C_{\alpha,c,\Delta s_t} \tau_\alpha(s_n)$) is applied to each point of the $x_2$-$x_3$-plane and it is not a function of the location of different points in the $x_2$-$x_3$-plane. Fig. 4 shows the resulting relationship between intrinsic time scale (s) and real-time scale (t) used in the time mapping process of the original Mann field to the new Time-mapped Mann field according to Eq. (19) for different values of $\alpha$ (in the Lévy distribution). As noticed from this figure, the relationship between $s$ and $t$ is not an exact straight line, which means that the resulting field will be different from the original field as explained in Sec. 3. The values of $\tau_\alpha$ in Eq. (19) follow a Lévy's distribution with the corresponding PDFs presented in Fig. 5 at different values of $\alpha$. In the following analysis and simulations, we will only consider $\alpha = 0.6$, which is highlighted with red lines in Fig. 4 and 5. Additionally, values of $c = 20$ and $\Delta s_t$=8 sec were considered for the calculation of the scaling factor $C_{\alpha,c,\Delta s_t}$ in Eq. (19). These parameters were selected through an iterative process to achieve values of Kurt($v_1$), defined in Eq. 11, in a comparable range to the data shown in Fig.1 and to other literature values like Schwarz (2020). The iteration aimed not only to find values of the parameters that match the desired values of Kurt($v_1$) but also to keep other statistical values such as the standard deviation and the spectrum of the time series of the wind velocity.

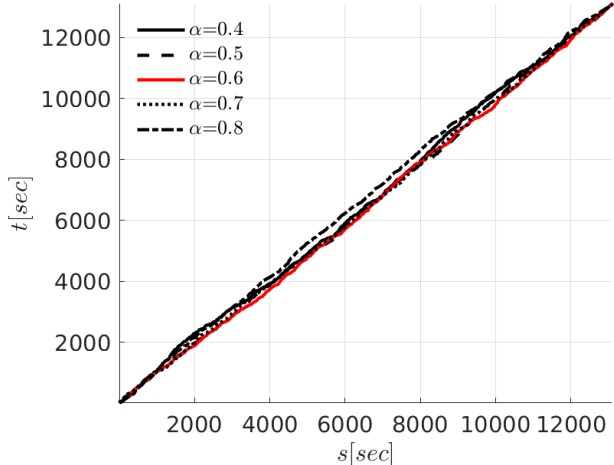

**Figure 4.** Relationship between intrinsic time scale (s) to physical time scale (t) (resulted from $\tau_\alpha$ in Eq. (19)) at different values of $\alpha$.

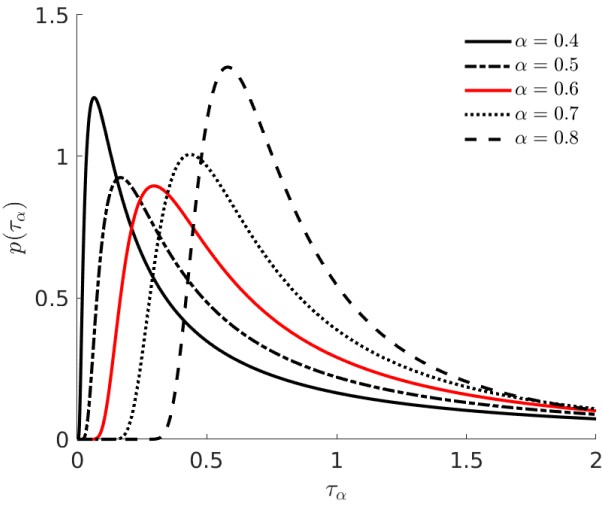

**Figure 5.** Lévy probability distribution of $\tau_\alpha$ as in Eq. (17) at different values of $\alpha$.

The comparison between the Mann and the new Time-mapped Mann wind fields regarding statistical properties is performed in terms of one-point and two-point statistics from Sec. 2.1. Starting with one-point statistics, the values of the mean wind speed in the longitudinal direction $U_1$ and the corresponding standard deviation $\sqrt{\langle u_1^2 \rangle}$ are presented in Table 2. These values are calculated from the time series at the hub-point (location of the hub of the turbine in the BEM simulations, see Sec. 4.2). For better comparison, the relative differences (Rel.Diff) calculated as the difference between both results over the result from

Mann field (i.e. (Time-map. − Mann)/Mann), are also presented in the table.

The values of the relative differences for $\langle U_1 \rangle$ and $\sqrt{\langle u_1^2 \rangle}$ are lower than 2%. This means, in terms of one-point statistics

**Table 2.** Values of the mean wind speed and standard deviation at the hub-point in the longitudinal direction ($\langle U_1 \rangle$ and $\sqrt{\langle u_1^2 \rangle}$) for the Mann (Mann) and Time-mapped Mann (Time-map.) wind fields.

| mean $\langle U_1 \rangle$ (hub) | | | standard deviation $\sqrt{\langle u_1^2 \rangle}$ (hub) | | |
|---|---|---|---|---|---|
| **Mann** | **Time-map.** | **Rel.Diff** | **Mann** | **Time-map.** | **Rel.Diff** |
| $20.02\,m/s$ | $19.98\,m/s$ | $-0.20\%$ | $1.99\,m/s$ | $1.96\,m/s$ | $-1.51\%$ |

at hub-point, the two generated wind fields are comparable. The slight differences between the two fields are caused by the interpolation procedure explained in Fig. 3. However, one-point statistics are not enough to show the effects of intermittency. To show the effect of the time mapping on the spectral properties of the new field, Fig. 6 shows a comparison between the spectra $\langle F_i(\kappa_i) \rangle_{\mathcal{I}}$ of three components of the original Mann field and the new Time-mapped Mann field where $\kappa_i = 2\pi f/U_i$. In these figures, $\langle . \rangle_{\mathcal{I}}$ denotes spatial averaging over the plane $\mathcal{I} = \{(x_2, x_3)|0 \leq x_2 \leq l_2, 0 \leq x_3 \leq l_3\}$. Fig. 6(a) shows the wavenumber spectrum of the $u_1$-velocity component in the longitudinal ($x_1$) direction, while Fig. 6(b) and 6(c) show the same comparison but for $u_2$ component in the $x_2$- and $u_3$ component in the $x_3$-directions, respectively. The theoretical spectra of the Mann field from Eq. (14) are plotted in the same figures to show how the spectra should be distributed over the different wavenumbers. Further to that, an uncorrelated, intermittent wind field was also added for the comparison. This uncorrelated wind field is generated by arbitrarily shuffling the grid points of the intermittent Time-mapped Mann field in the transverse directions ($x_2$- and $x_3$-directions). Of course, that still means that all points in the $x_2$-$x_3$-plane are shifted by the same time step in the temporal direction but now the spatial correlations in the $x_2$-$x_3$-directions are disturbed by mixing. This uncorrelated field is generated to show the difference between a correlated field in the transverse direction (which is the Time-mapped Mann field) and a completely uncorrelated turbulent field in the transverse direction.

Fig. 6 shows that both, the Time-mapped Mann field and the Mann field slightly differ from each other. Discrepancies arise through the time-mapping procedure and the interpolation procedure that is needed as described in Sec. 3. As indicated in Sec. 2.1, the velocity spectrum is not sufficient to show intermittency. Therefore, increment statistics should be used in this case to show the effects of intermittency. Fig. 6(a) shows that both, the Time-mapped Mann field and the Mann field slightly differ in the lower and higher wavenumbers whereas both are still close to the theoretical curve. Fig. 6(b) and (c) show that both, the Mann and Time-mapped Mann have the same behavior for wavenumbers starting from $\kappa_2 = \kappa_3 \approx 8 \cdot 10^{-2}$. Velocity spectra in $x_2$- and $x_3$-directions at wavenumbers $\kappa_2, \kappa_3 \lesssim 8 \cdot 10^{-2}$ are not plotted since the low number of grid points in $x_2$- and $x_3$-directions are too low to enable capturing spectra at such low wavenumbers using Fourier transform. The deviations between Mann and Time-mapped Mann fields happen due to interpolation errors that are more obvious in the transverse directions due to the low number of grid points, and accordingly a low number of interpolation points, in the transverse directions. It should be noticed that the velocity spectrum in Fig. 6(a) shows a noisy spectrum while the spectrum in figures 6(b) and (c) are smoothed. This happens due to the difference in the number of grid points in $x_1$-, $x_2$-, and $x_3$-directions. As indicated in

Tab. 1, $n_{x_1} = 131072$ while $n_{x_2} = n_{x_3} = 32$. This means that the number of points in $x_1$-direction for which the spectrum was obtained via Fourier transform is 4 orders of magnitude higher compared to the number of points in $x_2$- and $x_3$-directions.

As shown in Fig. 6, the spatial correlations in the streamwise direction are influenced by the time-mapping, while temporal intermittency could be successfully added. Since the velocities in the transverse directions are not shifted, there is no intermittency added in these directions. Furthermore, skewness is also not added to the model.

From turbulence theory, there exist many other models which address more statistical characteristics and are based on a proper superposition of Gaussian statistics (Rosales and Meneveau (2006)). A similar approach has also recently been applied to the 350 Mann model by a group from Oldenburg where the small-scale Gaussian statistics of Mann wind fields with varying covariances were superposed to yield small-scale intermittency (Friedrich et al. (2021)).

However, our intention was to combine well-established models in wind energy which we did by combining the Mann model with the time-mapping procedure from the CTRW model. As a consequence, our model shows spatial correlations in the $x_2$-$x_3$-plane similar to the Mann model and allows to investigate the influence of temporal intermittency.

The new model allows a detailed comparison between Gaussian Mann fields and non-Gaussian Time-mapped Mann fields. This comparison is shown in detail in this work. For further investigation of these deviations between Mann and Time-mapped Mann fields, Fig. A1 shows the same spectra as shown in Fig. 6 but with an equal number of grid points in all directions (in this case, $n_{x_1} = n_{x_2} = n_{x_3} = 512$). This case will be referred to as the "cubic case". The spectra of these fields in Fig. A1 show that for a finer discretization in $x_2$- and $x_3$-directions, the discrepancies between the curves almost vanish, while the most obvious 360 deviation between the two cases can be seen in Fig. A1(a). This case can show the effect of the number of grid points and the direction on the spectra of the Mann and Time-mapped Mann field since it has more grid points in $x_2$- and $x_3$-directions than the former case shown in Fig. 6.

For the uncorrelated intermittent field, a comparable agreement as in the correlated case between this field and the Time-mapped Mann field in the longitudinal direction $x_1$ can be seen since the arbitrary shuffling was only applied in the transverse 365 directions. For the spectra in the transverse directions, The blue dashed lines were removed since $\langle F_2(\kappa_2) \rangle_\mathcal{I}$ and $\langle F_3(\kappa_3) \rangle_\mathcal{I}$ values are around unity regardless of the wavenumber, which proves the destruction of the correlation in the transverse directions due to this shuffling.

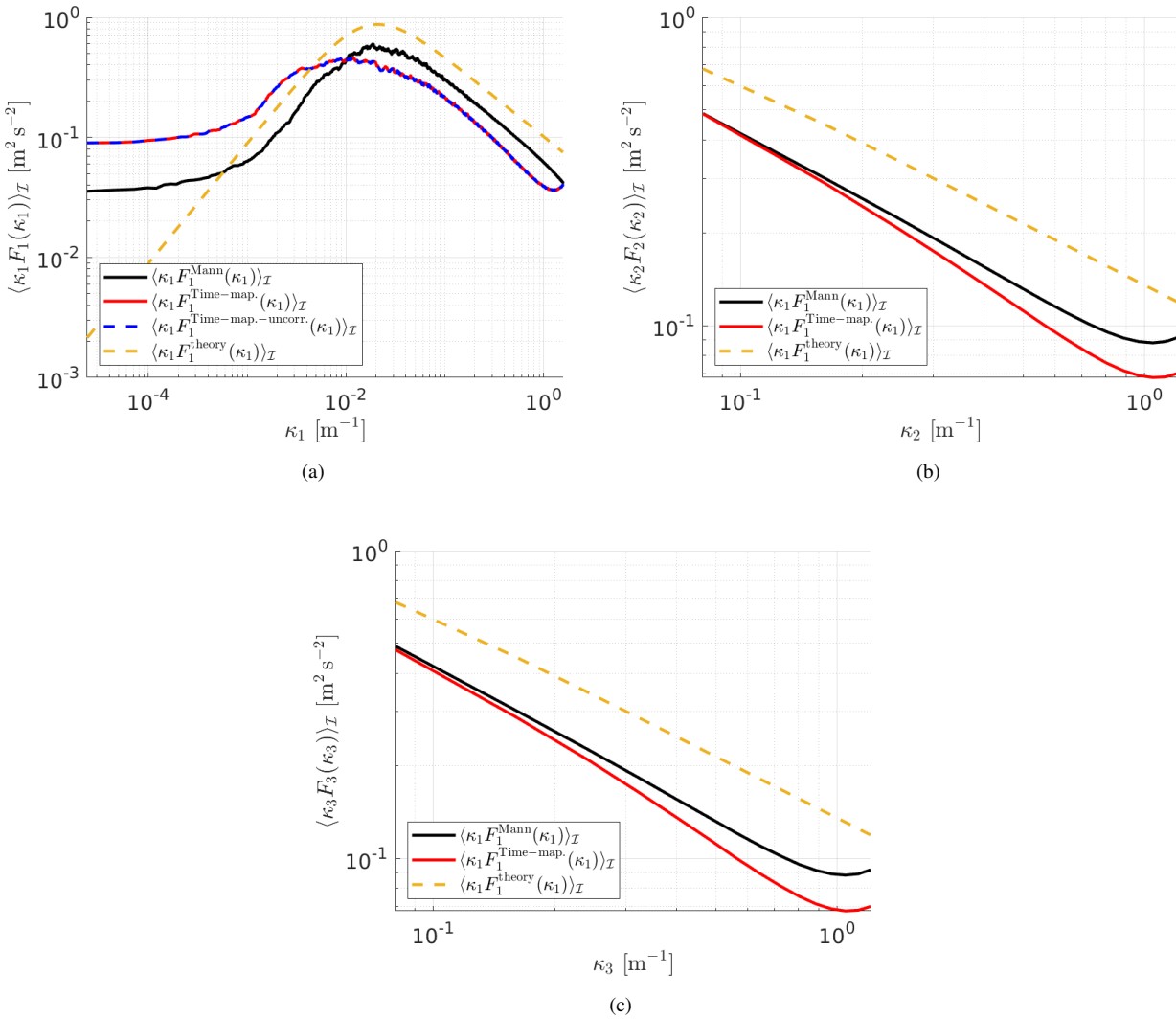

(a)

(b)

(c)

**Figure 6.** Comparison of averaged spectra in (a) $x_1$-, (b) $x_2$-, and (c) $x_3$-directions for the Mann field ($F_i^{\mathrm{Mann}}(\kappa_i)$), the Time-mapped Mann field ($F_i^{\mathrm{Time-map.}}(\kappa_i)$), an uncorrelated Time-mapped Mann field e.g.: ($F_i^{\mathrm{Time-map.-uncorr.}}(\kappa_i)$) and the theoretical spectra ($F_i^{\mathrm{theory}}(\kappa_i)$) calculated from Eq. (14)

In the comparison between Mann and Time-mapped Mann fields, the velocity spectra in $x_2$- and $x_3$-direction are expected to be similar since these one-point spectra are not able to show the effect of the time mapping. Therefore, to see the differences between the Mann and the Time-mapped Mann field, the coherence of velocity components (Eq. (6)) of the two wind fields in the three spatial directions is plotted versus $\kappa_1 \Delta x_2$ in Fig. 7 with $\Delta x_2 = 2.6m$. In this figure, the difference in the coherence of the two fields as we move in $x_2$-direction are obvious due to the time mapping in the longitudinal direction. The Mann model is based on the von Kármán energy spectrum, which is reflected by the coherence of the Mann-modelled wind field. On the other

370

hand, a comparison between the coherence of the two wind fields versus $\kappa_2 \Delta x_3$ for the cubic case is shown in Fig. A2 in the appendix. In the latter case, the differences in coherence between the two fields are small. This is due to the unchanged $x_2$-$x_3$ planes during the time mapping process. This shows that the coherence with respect to the longitudinal direction is highly influenced by the time mapping whereas it is not with respect to the transverse direction. That shows, the time-mapping goes at the cost of the spatial correlations in $x_1$-direction which is expected since the $x_1$-axis is stretched and compressed, which is also reflected by the spectrum in $x_1$-direction in figure 6(a). By comparing Fig. 7 and Fig. A2, it can be noticed that the plots in Fig. A2 look smoother. This is also due to the difference in the number of grid points in the $x_1$-direction between the two cases.

Following on from the two-point statistics analysis, the velocity increments $v_1$ defined in Eq. (10) and the degree of intermittency of the wind fields are analyzed. Fig. 8 shows the comparison between the PDFs of the velocity increment of both, the Mann and the new Time-mapped Mann wind fields. It is obvious in this figure how extreme events in the case of the Time-mapped wind field (the red markers) have a higher probability than the corresponding events in the case of the Gaussian Mann field (the black markers). Also, it can be noticed that the intermittency increases with the decrease of $\tau$. Note that this $\tau$ refers to the temporal increment in Eq. (10) and not to the $\tau_\alpha$ introduced in the context of the Lévy distribution. Another important measure of the increment statistics is the kurtosis calculated from Eq. (11). In the case of the kurtosis of the wind fields, the angular brackets denote averaging over time and kurtosis, in this case, is only a function of $\tau$ Fig. 9 shows a comparison between the kurtosis of the velocity increment from the Mann field and the Time-mapped Mann fields at different $\tau$ values. As explained earlier, higher values of $\tau$ show a Gaussian distribution with kurtosis of 3 and lower values of $\tau$ show more deviation from the Gaussian PDF of the Mann field.

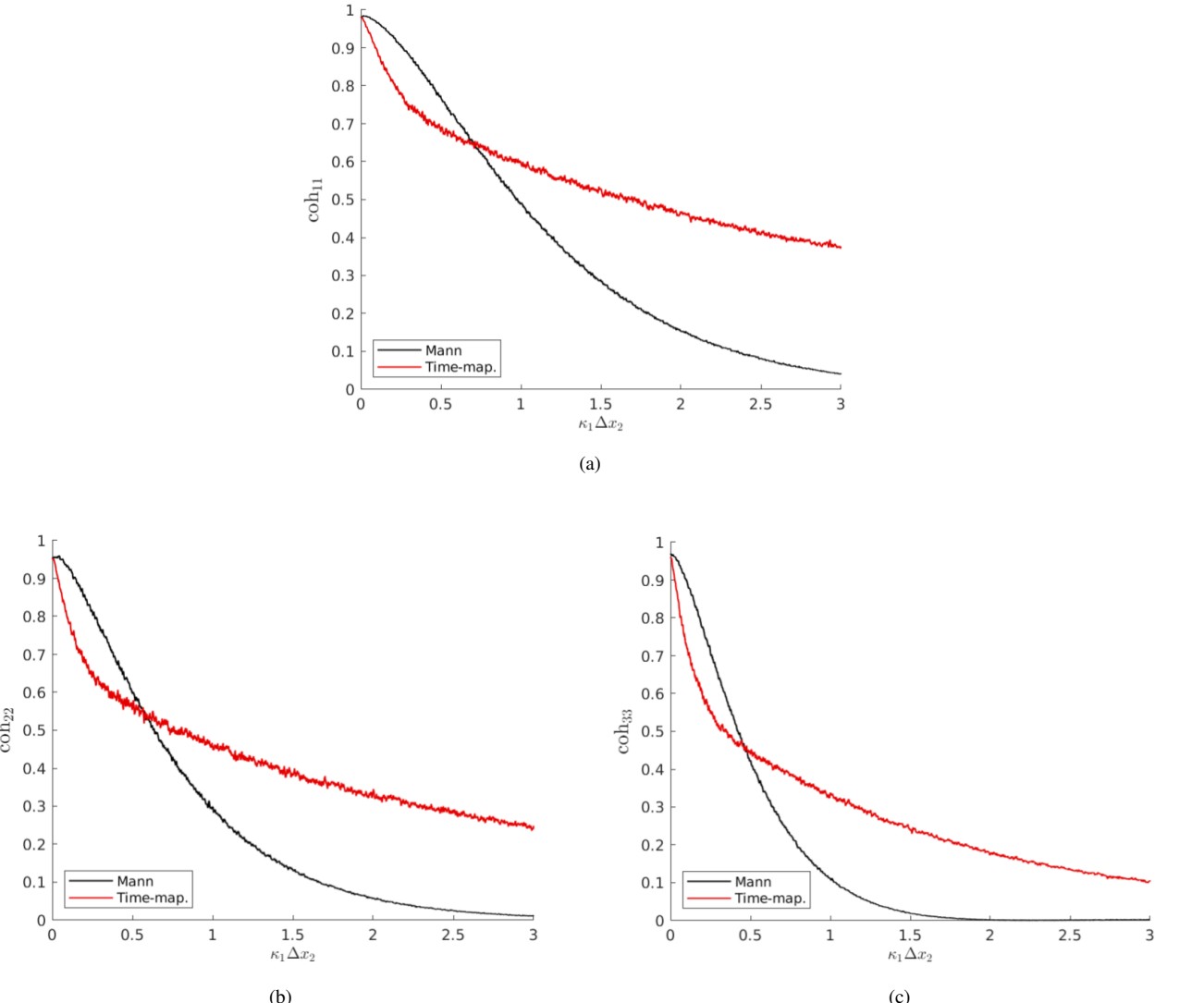

**Figure 7.** Comparison of the coherence of velocity components in (a) $x_1$-, (b) $x_2$-, and (c) $x_3$-directions for both Mann and Time-mapped Mann fields over wavenumber $\kappa_1 \Delta x_2$ with $\Delta x_2 = 2.6$m and $\Delta x_3 = 0$. Dark lines represent smoothed coherence values of the light lines using a moving average method.

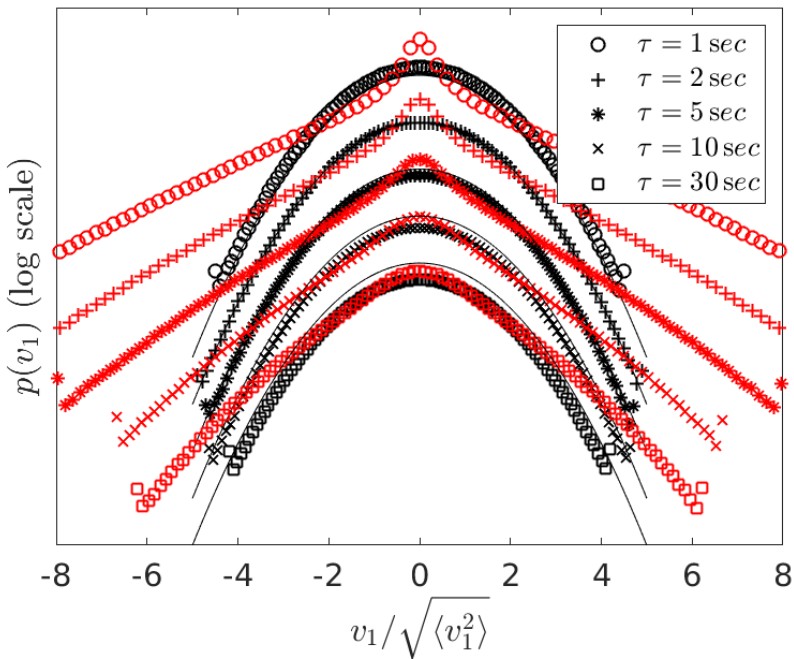

**Figure 8.** Spatially averaged PDF over transverse planes of velocity increments of the Mann wind field (black) and the Time-mapped Mann wind field (red) at different temporal increments $\tau$. The black thin lines are Gaussian distributions as a reference. In principle, the curves would all lie on top of each other but here, the curves are shifted vertically for a better visualization.

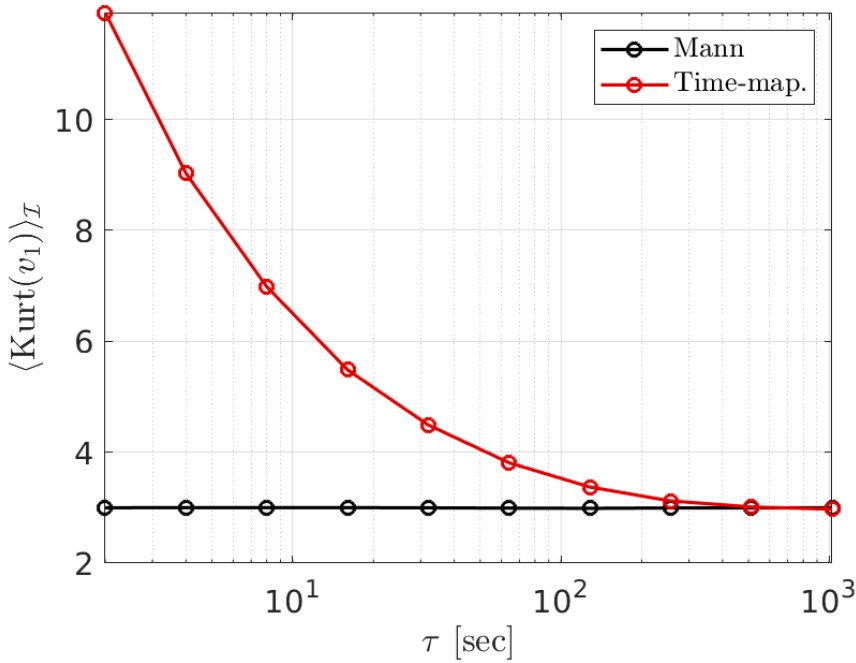

**Figure 9.** Kurtosis of Mann (black) and Time-mapped Mann (red) fields (Eq. (11)) at different temporal increments $\tau$. For a Gaussian distribution the kurtosis is 3.

## 4.2 Analysis of the resulting wind turbine loads

After analyzing the effect of the time mapping on the different statistics of the wind fields in the previous section, here, the
effect on exemplary turbine loads is studied. The main point of this section is to see whether the intermittency from the wind field is carried to the turbine structure. If the turbine loads are proved to reflect intermittency, this might have a significant effect on the dynamic loads acting on the wind turbine. For example, the damage equivalent loads (DEL) calculated from fatigue analysis might be affected due to intermittency because the probability for extreme values in the load increments is larger than for a Gaussian distribution.
In this work, BEM simulations of the NREL WindPACT 1.5MW (Malcolm and Hansen (2006)) virtual wind turbine are performed using the aero-elastic simulator NREL FAST (v8.16) (Jonkman and Jonkman (2016)). The main characteristics of the turbine and input parameters for the simulations are summarized in Table 3. The wind fields used in these simulations are the same Mann and Time-mapped Mann fields analyzed and compared in Sec. 4.1.

| Parameter | Value/Description |
|---|---|
| Mean wind speed $\langle U_1 \rangle$ | 20 $m/s$ |
| Mean wind speed $\langle U_2 \rangle$ and $\langle U_3 \rangle$ (averaged over transverse plane) | 0 $m/s$ |
| Turbulence Intensity ($TI_1 = TI_2 = TI_3$) at hub (Eq. (2)) | 0.10 |
| Wind shear | No |
| Rotor diameter | 70 m |
| Hub height | 84 m |
| Control | Fixed pitch - Fixed speed |
| Rotor speed | 20 rpm |
| Pitch angle | 20° |
| Blades structural simulation | Rigid |
| Simulation time | 13100 sec |
| Sampling frequency | 20 Hz |
| Airfoil aerodynamics | Unsteady - (Leishman and Beddoes (1989)) |
| Tip and hub-loss | Prandtl |
| Rotor tilt | 5° |
| Gravity effects | Turned on |
| Structural degrees of freedom (DOF) | Turned off |
| Tower passage | Turned off |
| Wake model | Induction-BEM Model |

**Table 3.** Main parameters of the turbine and numerical simulations. A description of the listed set-ups for the BEM simulation can be found in Jonkman et al. (2016) and Jonkman and Jonkman (2016).

Four different load sensors were selected for analyzing the effect of the intermittent wind: the bending moment at the root of the blade in the flapwise direction (`RootFlap`), the rotor torque (`Torque`), the rotor thrust (`Thrust`) and the bending moment at the base of the tower in the fore-aft direction (`TwrForeAft`). According to Schwarz (2020), these four sensors are expected to be highly sensitive and mainly dominated by aerodynamic forces in the direction of the flow and not by other sources of load such as gravitational forces.

Firstly, the mean values and standard deviations calculated over the length of the simulation are presented in Table 4. Here, the results for the four load sensors of both, the Time-mapped Mann (Time-map.) and the Mann (Mann) wind field are shown. Moreover, the relative differences (Rel.Diff) were also calculated (i.e. (Time-map. $-$ Mann)/Mann). In Table 4, the relative differences of the mean values and standard deviations of the analyzed sensors are less than $\pm 1\%$ between the Mann and the Time-mapped Mann wind field. It can be seen that this difference is in the same order as the differences in the mean wind

speed $\langle U_1 \rangle$ and standard deviation $\sqrt{\langle u_1^2 \rangle}$ at the hub point presented in Table 2. To investigate whether the intermittency in the wind field carries over to the turbine loads, the incremental statistics of the loads are evaluated. The load increments $M_\tau$ are

| Sensor/Load | Mean Value | | | Standard Deviation | | |
|---|---|---|---|---|---|---|
| | Mann | Time-map. | Rel.Diff | Mann | Time-map. | Rel.Diff |
| RootFlap | 861 kN.m | 868 kN.m | 0.81% | 253 kN.m | 251 kN.m | −0.79% |
| Torque | 1126 kN.m | 1136 kN.m | 0.89% | 269 kN.m | 267 kN.m | −0.74% |
| Thrust | 157.76 kN | 158.77 kN | 0.64% | 24.16 kN | 24.09 kN | −0.29% |
| TwrForeAft | 10317 kN.m | 10406 kN.m | 0.86% | 2017 kN.m | 2017 kN.m | 0% |

**Table 4.** Mean values and standard deviations of time series from BEM simulations for the Mann (Mann) and the Time-mapped Mann (Time-map.) wind fields. The relative differences (Rel.Diff) between the two cases are calculated for better interpretation.

defined similarly to velocity increments in Eq. (10):

$$M_\tau(t,\tau) = M(t+\tau) - M(t) \tag{21}$$

where $M$ is the time series of the load sensor and $\tau$ corresponds to the time lag. Fig. 10 shows the PDFs of $M_\tau$ for the four selected load sensors, for $\tau = 3$s (top) up to $\tau = 30$s (bottom) for the blade root flapwise bending moment and $\tau = 1$s (top) up to $\tau = 30$s for the rotor thrust, rotor torque, and tower foreaft bending moment. For a direct comparison to the PDFs of the velocity increments $v_1$, the values of $\tau$ are selected analogously to Fig. 8. Similar to Fig. 1 and 8, all the indivual PDFs are normalized to standard deviation $\sqrt{\langle M_\tau^2 \rangle} = 1$. Here, the angular brackets denote moving averages over the time series.

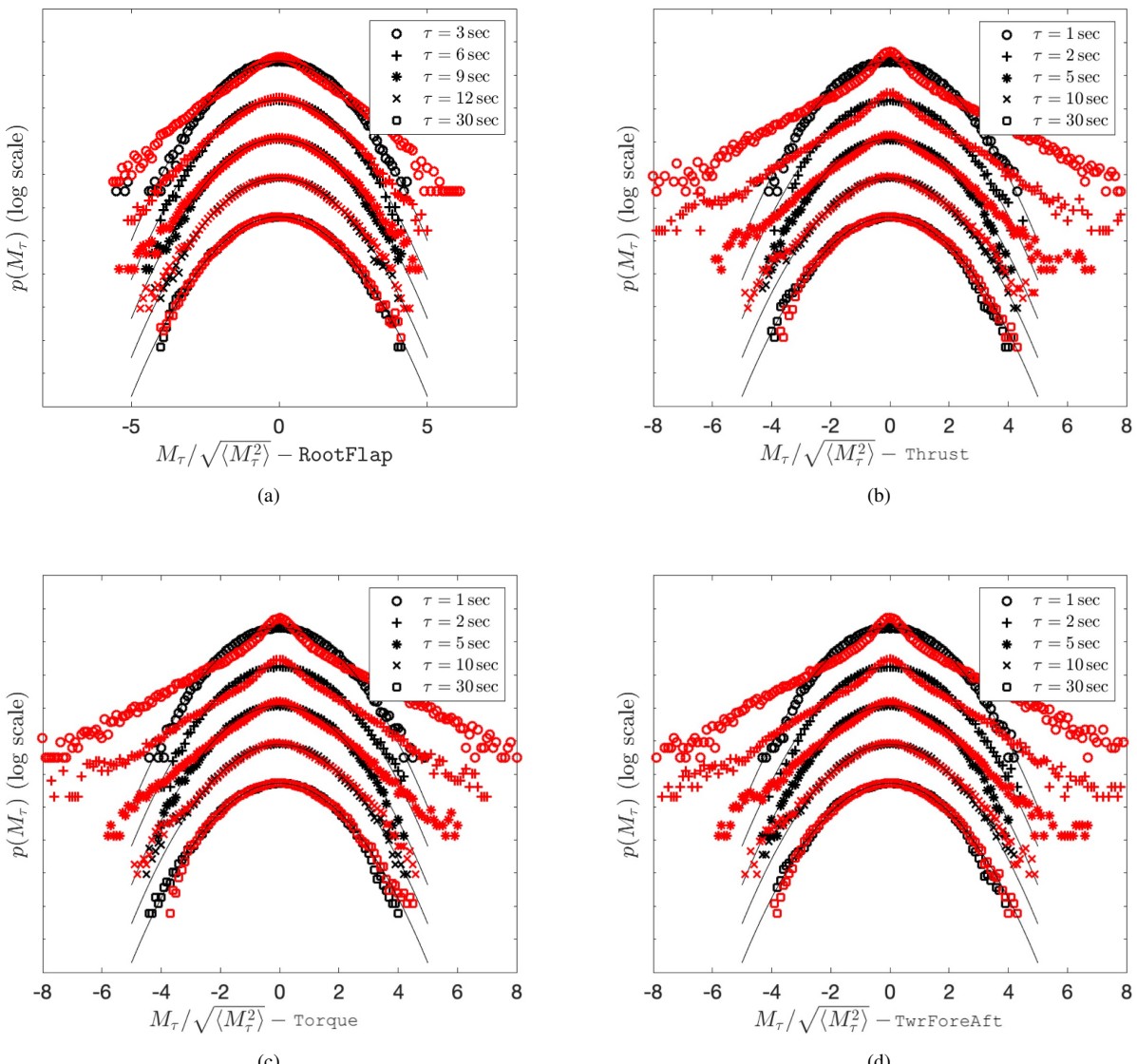

**Figure 10.** PDFs of load increments $M_\tau$ for (a) blade root flapwise bending moment (`RootFlap`) (b) rotor thrust (`Thrust`) (c) rotor torque (`Torque`) (d) tower fore-aft bending moment (`TwrForeAft`) of the 1.5MW turbine with Mann and Time-mapped Mann inflow wind fields at different temporal increments $\tau$. The red markers correspond to the Time-mapped Mann field cases while black markers present the results for the Mann field cases. The black thin lines are Gaussian distributions as reference. The curves are shifted vertically for visualization.

The resulting load increments $M_\tau$, follow the characteristics of the wind velocity increments shown in Fig. 8. The PDFs of the Mann load increments are very well described by the Gaussian distribution. On the contrary, the PDFs of the load increments resulting from the Time-mapped Mann field deviate significantly from the Gaussian distribution for a range of scales.

The observed heavy tails shown by the Time-mapped PDFs reflect, as mentioned earlier, a higher probability of extreme load fluctuations than the probability of a Gaussian distribution.

Fig.10(a) shows a special case for the `RootFlap` load. The shown $\tau$-values differ from the other loads to make the intermittency visible in this case. The `RootFlap` moment is dominated by the 1P frequency while the other loads are dominated by the 3P frequency. Consequently, showing the same $\tau$-values in Fig.10(a) as for the other loads would not reveal the intermittency in our special case. The energy spectra of the `RootFlap` and `Thrust` are shown in Appendix B for a deeper understanding of the different frequencies of the loads.

Also, It turns out that the frequencies of the energy spectra carry over to the frequencies in the kurtosis, as shown in Fig.11 for the thrust. A more detailed description of the evolution of the intermittency of $M_\tau$ with $\tau$ is presented in Fig. 11. Here, the results of $\mathrm{Kurt}(M_\tau)$ from Eq. (11) are calculated exemplarily for the `Thrust` signal. It is visible that the values of the kurtosis for the load case with the Time-mapped Mann wind field are higher than three (the value of three indicates a Gaussian distribution) for all the values of $\tau$ up to around 20 seconds. This proves the intermittent characteristics of the loading. In

contrast, the corresponding values for the Mann case fluctuate around three for all the considered time scales, agreeing with the Gaussian statistics. Additionally, for time scales below one second, a decreasing intermittent behavior of the load is visible in the Time-mapped case when decreasing $\tau$. The kurtosis of the thrust in Fig 11 is overall lower than of the corresponding wind field as shown in Fig 8. This has been also shown by Mücke et al. (2011) who have used an atmospheric wind measurement as input to calculate the torque.

Further investigations should be done to explain the potential sources of the observable bumps in the kurtosis of the Time-mapped Mann case. A strong dependence of the peaks on the rotational frequency of the rotor has been recognized by the authors. However, this first turbine study aimed at investigating whether the intermittency which was introduced to the wind field by deriving the Time-mapped Mann field carries over to turbine loads in general. This has clearly been shown here.

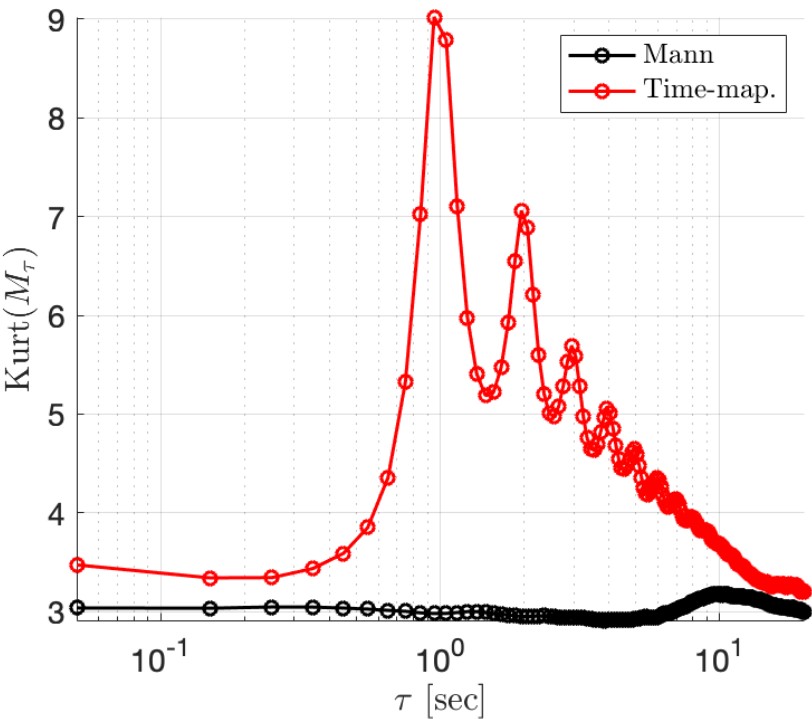

**Figure 11.** Kurtosis of the PDF of $M_\tau$ calculated for the `Thrust` according to Eq. (11) at different temporal increments $\tau$. The red markers correspond to the Time-mapped Mann field while black markers present the results for the Mann field. A value of three corresponds to a Gaussian distribution.

## 5  Conclusions

In this work, a new synthetic wind field model has been introduced which combines spatial correlations from the well-known Mann model and the effect of intermittency. During the derivation, the time-mapping procedure from the so-called CTRW model has been applied to a Mann wind field to generate intermittency. This new model is called the Time-mapped Mann model. The CTRW model relies on stochastic differential equations for the velocity and assumes exponentially decaying velocity correlations. On the other hand, the Mann model is based on a proper modeling of the spectrum according to atmospheric

conditions which are also recommended by the standard IEC 61400-1. With our procedure, we obtain a model which aims at keeping the spatial correlations in the transverse directions from the Mann model, and thus, following the IEC 61400-1 standard in this respect, and on the other hand adding intermittency which has also been reported as a feature inherent in the wind (Mücke et al. (2011)).

The analysis of the velocity spectra, as well as the PDF and kurtosis of velocity increments, showed that this method managed

to generate an intermittent wind field in the longitudinal direction. On the other hand, this method keeps the spatial correlation

almost exactly as in the Mann field in the other two directions. This is due to the way the time mapping is applied where the velocity field "slices" ($x_2$-$x_3$ planes) of the Mann field are shifted by random time shifts. Small differences in the compared statistics in transverse directions are due to the interpolation between the generated time-shifted slices and the new, uniformly separated velocity field slices.

Furthermore, the first analysis and comparison of loads with respect to intermittency between Mann and Time-mapped wind fields on a 1.5MW wind turbine was done. The analysis of the loads showed that the intermittency is transported from the wind field to the structure of the wind turbine. However, these loads are different in their response to the wind field intermittency. The increment statistics showed that in the analyzed case, rotor thrust, rotor torque, and tower fore-aft bending moment are strongly affected by the intermittency in the wind field, whereas the blade root bending moment shows less intermittency. The

resulting non-Gaussian load increment distributions suggest that simulating wind turbines under Gaussian wind fields such as Mann wind fields could potentially lead to an underestimation of the probability of extreme loads.

An important future extension of the Time-mapped Mann model would be an incorporation of shear in the Time-mapped Mann model. The original Mann model has the option of imposing eddy stretching to account for shear. However, it may not be a straightforward process to add this eddy stretching to the Time-mapped Mann field. The first question is whether the eddy

stretching should be added before or after the time-mapping, whereas the option to add it before the time-mapping process may seem more reasonable. Open remains, however, the effect on the eddy stretching since the time-mapping also introduces a similar stretching effect. Further, it has to be investigated if the areas with larger mean velocity should have a different time-mapping.

Also, in future research based on this work the different turbine loads and their response to turbulent inflow with intermittency

could be investigated in more detail. This investigation should involve also different sizes of wind turbines that cover a larger area of the wind fields, as well as variable speed, variable pitch wind turbines at a variety of wind conditions. For example, such wind conditions could be measured from wind sites to have better anticipation of the turbine loads in the presence of intermittency. Another possible future work would be to compare the impact of intermittency on different wind turbine loads at different wind speeds to investigate whether there are effects of wind speeds on the loads' intermittency. Additionally, fatigue

load could be a subject of further investigation which requires further discussion since the rainflow cycle counting method may not detect intermittency properly as discussed by Mücke et al. (2011).

## Appendix A: Analysis of a cubic field

For a further investigation of the effect of the time mapping of wind fields on the spectra, a Mann box with $n_{x_1} = n_{x_2} = n_{x_3} = 512$ and $\Delta x_{1,m} = \Delta x_{2,m} = \Delta x_{3,m} = 1m$ is studied in this appendix. The same comparison of spectra as described in Sec. 4.1

is studied here for this cubic field. As introduced in the previous sections, there are three different parameters used to generate the box: the overall box size, the grid cell size, and the number of cells. Since it is not possible to change the value of one of these parameters alone without changing at least one of the other two parameters to generate a grid with comparable results, a new grid with cubic cells was generated to have independence of the direction since all directions have the same grid points

and cell sizes.

In Fig. A1 (a), (b), and (c) it can be seen that the spectra show generally good agreement between the Mann and the Time-mapped Mann field. The spectra in the longitudinal direction $x_1$ in A1(a) show a discrepancy due to the time-mapping procedure which is applied to this direction. Further, all plots show increasing discrepancies for larger wavenumbers which should be resulting of the interpolation procedure. Note that in Fig. 6(b) and (c) where the resolution in $x_2$- and $x_3$-direction was much coarser, the deviations already start at smaller wavenumbers. The comparison of coherence of the same two fields in Fig. A2

shows good agreement between the two fields in the $x_2$- and $x_3$-directions. The small deviations result from the interpolation routine.

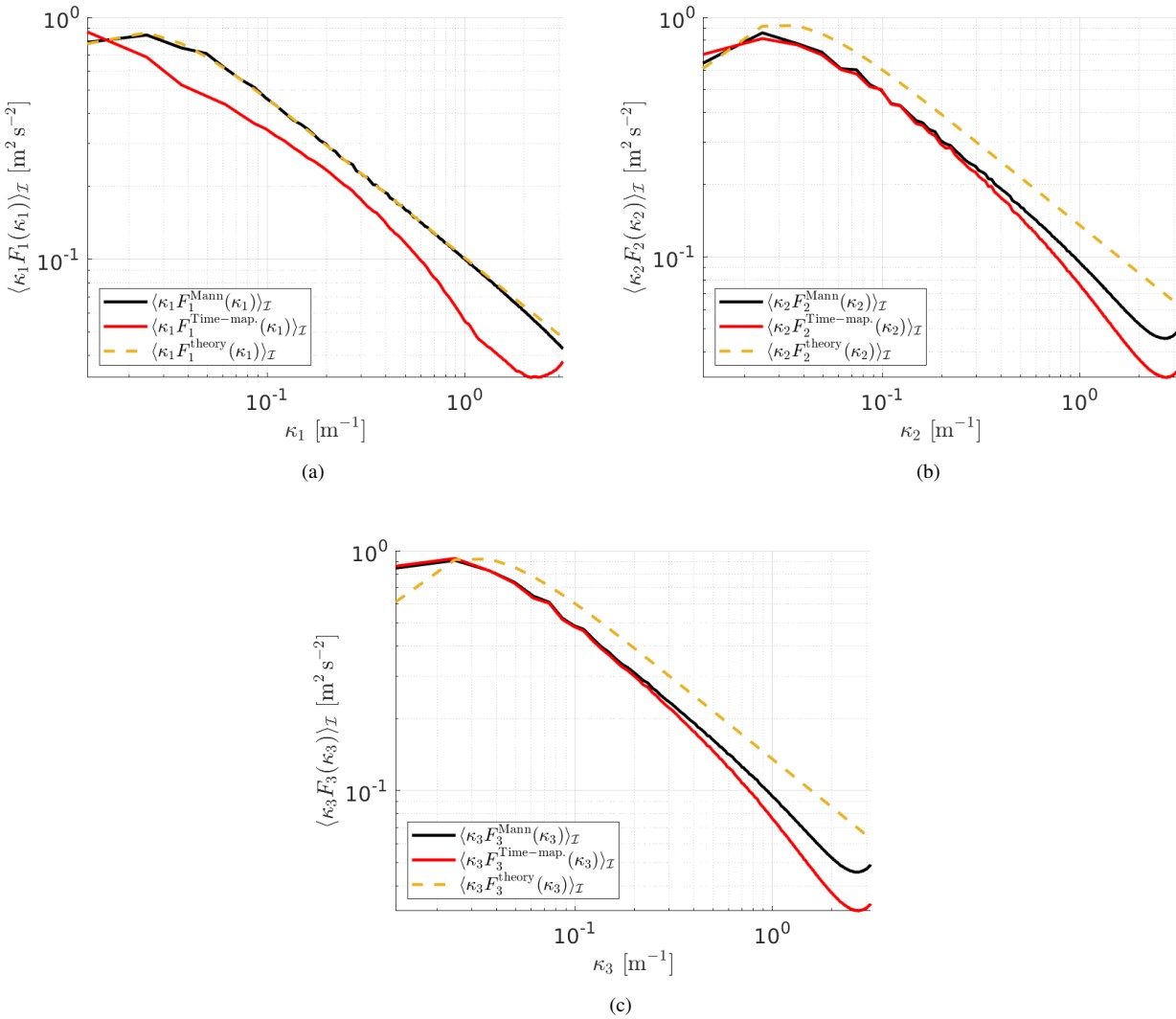

**Figure A1.** Comparison of averaged spectra in (a) $x_1$-, (b) $x_2$-, and (c) $x_3$-directions respectively for the Mann field, the Time-mapped Mann field and the theoretical results calculated from Eq. (14)

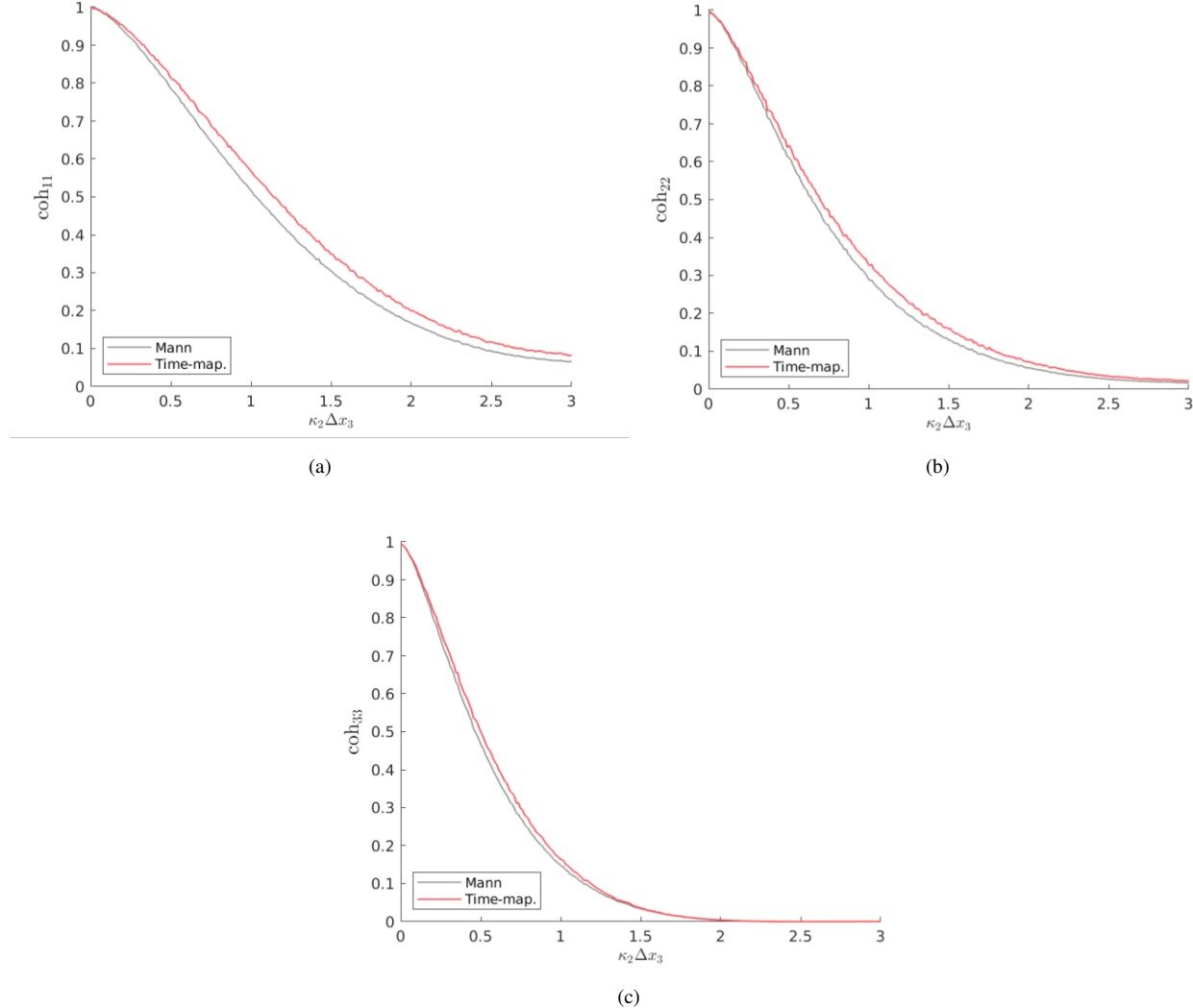

(a)

(b)

(c)

**Figure A2.** Comparison of the coherence of velocity components in (a) $x_1$-, (b) $x_2$-, and (c) $x_3$-directions for the Mann field and the Time-mapped Mann field over wavenumber $\kappa_2 \Delta x_3$ with $\Delta x_3 = 2.6$m and $\Delta x_1 = 0$.

## Appendix B: Analysis of the energy spectra of different loads

To investigate the phenomenon shown in Fig. 10(a), in which the `RootFlap` shows intermittency at other $\tau$ values, Fig. B1 shows a comparison between the energy spectra of two different loads, namely the blade root flapwise bending moment `RootFlap` and the rotor thrust `Thrust`. It can be seen from this figure that the energy spectrum of the `Thrust` shows peaks at the 3P frequencies and higher whereas the blade root flapwise bending moment `RootFlap` shows peaks at the 1P frequency and higher.

This indicates that the fluctuations of the loads happen at different frequencies, which entails applying different $\tau$ values in the PDF to be able to see the intermittency as shown in Fig. 10.

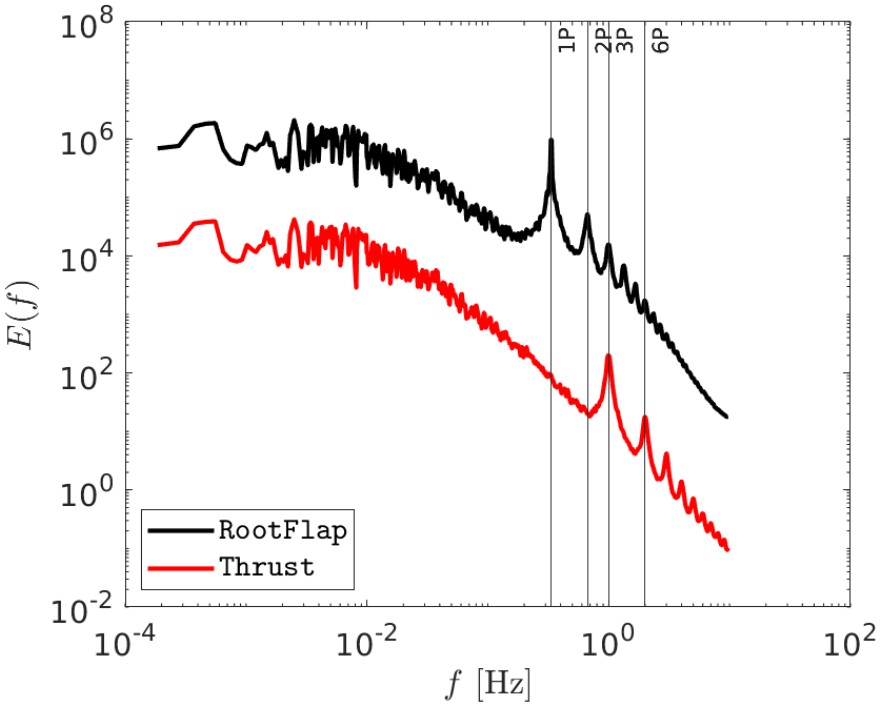

**Figure B1.** A comparison between the energy spectra of the blade root flapwise bending moment `RootFlap` and rotor thrust `Thrust` of the simulated wind turbine under a turbulent wind field generated by the Time-mapped Mann model.

*Author contributions.*

KY performed the wind field simulations and the wind field analyses and wrote the paper (with contributions from the other authors). Preliminary evaluations were performed in the scope of the Master Thesis of AH at the Carl von Ossietzky University Oldenburg supervised by LJL and LH. All authors contributed to the generation of the method. DM performed the BEM simulations and the load analysis. HK and LH provided intensive reviews on the generation of the codes and the analyses. LJL initiated the research, provided extensive consultation on the development of the method and the scientific analyses, and had a supervising function.

*Competing interests.* The authors declare no conflict of interest.

*Disclaimer.* TEXT

*Acknowledgements.* The authors would like to acknowledge Mr. Jörg Schwarte, Nordex Energy SE & Co.KG for providing measured
wind data. Parts of this work have been performed in the scope of the project EMUwind (03EE2031A/C), PASTA (03EE2024A/B), and
SFB1463 (434502799). The EMUwind and PASTA projects are funded by the German Federal Ministry for Economic Affairs and Energy
(Bundesministeriums für Wirtschaft und Energie-BMWi) due to a decision of the German Bundestag and the SFB1463 is funded by the
German Research Foundation (Deutsche Forschungsgemeinschaft, DFG).

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
