# Peer review of "Applying a Random Time Mapping to Mann modelled turbulence for the generation of intermittent wind fields"

_Wind Energy Science, 2021_

## Referee Comment (RC2)

Review of the manuscript:

*Khaled Yassin, Arne Helms, Daniela Moreno, Hassan Kassem, Leo Höning, and Laura J. Lukassen:*

**"Applying a Random Time Mapping to Mann modelled turbulence for the generation of intermittent wind fields"**

**General comments:**

The intermittency of atmospheric wind fields has already been examined in many studies in the past. The working group in Oldenburg has been working for a long time on the investigation of the influences of intermittent wind fields on the aerodynamic properties of airfoils and wind turbines and the development of methods for the synthetic generation of wind fields taking intermittency into account. The present manuscript represents a further contribution to this topic. It proposes the combination of the CTRW-based approach suggested by Kleinhans with the established Mann model for turbulence generation, in order to utilize the spatial correlations considered in the latter model. The differences to the original Mann model are exemplified by a generated wind field and the load statistics for a 1.5 MW turbine calculated with BEM by means of FAST. It could be shown that mean values and variances of the wind fields and loads agree well. However, the extended approach, denoted time-mapped Mann model, shows intermittencies, both, in the generated wind field and in the calculated loads except for the flapwise blade root bending moment. The consideration of shear, as foreseen in the Mann model, obviously requires further, considerable development effort in the CTRW approach and is still pending. The present results are therefore limited to a wind field without shear.

The manuscript is concisely written and contains a good and broad overview of the specific state of research. However, unconventional nomenclature, partial lack of introduction of variables in the text and inconsistent use of variables complicate readability. The results show some peculiarities in the predicted loads that may be attributed to the turbine BEM model rather than to the wind field model. The turbine model should be checked prior to interpretation of the results. In the results section, the authors largely limit themselves to describing the results visible in the pictures without analysing and explaining peculiarities in the results. In this respect the manuscript should definitely be improved. Moreover, I don't see why an Appendix is necessary and suggest to incorporate the content directly in the paper.

In conclusion the study is relevant and I in principle support publication of the manuscript if the following comments are taken into account and the interpretation of the results is elaborated.

**Specific comments and remarks:**

- Please introduce new properties throughout the document (like e.g. r in L. 104, $\Delta x_2$ and $\Delta x_3$ in L. 115, …) and use the variables consistently.
- L. 112: „where the double index does not indicate a summation here" is more confusing than helpful.
- L.132: Are $u_1$ and $u_2$ velocity fluctuations or absolute velocities? Compare eq. (1) small vs. capital letter.

- L. 138f.: An increasing deviation for reduced τ is hardly visible in Fig. 1 due to the vertical shift. Perhaps this can be made visible by a more appropriate choice of the ranges shown.

- L172: "different" → "multiple"/"several"

- Figure 2: To use $x_2$ as vertical component and $x_3$ as lateral is unusual (cf e.g. Mann, 1998). Please ensure consistency throughout the paper. N as well as the $\Delta$ values likely refer to the Mann box parameters. These are not introduced yet, please do so. Using $\Delta x_2$ and $\Delta x_3$ with a different meaning than in equation (6) confuses. I suggest to use e.g. $\Delta x_{1,M}$ and $\Delta x_{2,M}$ here.

- L. 255ff: Is there a specific reason why you prefer Eq. (21) instead the derivation according to IEC 61400-01 if no shear is considered? $L=0.8*\Lambda$ with $\Lambda=42m$ due to $z_{hub}>60m$ → $L=33.6m$

- Table 1: Please justify the choice of the Mann box parameter. How is length $n_{x1}$ chosen? $n*\Delta x$ in lateral direction is not even twice the integral length scale L; is this sufficient to resolve the statistics?

- L. 269ff: As Kleinhans pointed out, one disadvantage of the CTRW approach is the definition of the various parameters ($\alpha$, c, $\Delta s_t$) that control the intermittency. The general validity of the approach is limited in particular by the fact that $\alpha$ depends on the time lag and must be calibrated in order to correctly represent real intermittent wind fields. The values chosen in the present study do not contain a $\alpha(\tau)$ dependency and differ significantly for c, $\Delta s_t$ but also for $\alpha$ from the values chosen by Kleinhans. Are there any recommendations for selecting these parameters for practical cases and can the authors reason their choice?

- L. 279: Repetition of the statement from L. 276f.

- L. 283: Please include your definition of $\kappa$ ($=2*\pi*f/U$) and specify U to make sure whether it's an angular wavenumber.

- Fig. 6: Fig. (a) obviously shows a narrowband spectrum whereas (b) and (c) seem to be smoothed or have large frequency bins. Please justify this inconsistency or align.

- L 291ff: Please rework interpretation of Fig 6 and explain what spectra characteristic was to be expected for the new wind field model (e.g. L. 293: Does the difference between Time-mapped and original Mann in 6(a) for low wavenumber represent intermittency?);

- L. 294f: The statement cannot be followed because the small wavenumber range is not shown in plots 6(b) and (c).

- L296: The number of grid points in vertical and lateral direction is the same for the Mann model and the time-mapped Mann model, Therefore I cannot follow your explanation that the differences result from the grid. Please clarify.

- L. 297ff: The conclusion from comparison to A1 is not obvious. Ranges are different and therefore deviations are hardly comparable. Moreover, changes in grid size and resolution are mixed and hence effects cannot be attributed to a finer discretization for sure. It is still unclear why the size or resolution should explain differences between time-mapped and original Mann model.

- L. 308ff: Please mention the "settings" ($\Delta x_2$, $\Delta x_3$) also in the text and ensure that this distance values are not confused with Mann box parameters and elaborate the interpretation.

- L. 311ff: Can the authors please interpret this characteristic in more detail and explain the difference to the behavior for the cubic case in Appendix? The plots in A2 are somehow smoothed, please get it consistent with Fig. 7.

- Fig 7: Please do not mix (x,y,z) and $(x_1,x_2,x_3)$ and get it consistent with Fig. 2.

- P. 19, Tab. 3: I assume that the Dynamic Stall and the Dynamic Inflow models were activated in the FAST calculations. To be on the safe side, this should be mentioned. Moreover, does the turbine have tilt?

- Fig. 10: For better differentiation of the figures, the specific moment (RootFlap, ...) depicted in (a) to (d) could be added to the axis labels or the respective figures.

- L. 357ff: Fig. 10 shows interesting results. Fig. 10 (b) - (d) clearly show the influence of intermittency. However, Fig. 10 (a) does not show any impact of intermittency in the results with the new model. This is strange, since I cannot see how the blade bending loads and the thrust (as the sum of the three blade loads in the same direction) differ in their intermittent behavior that strongly. Please conduct further investigations to explain this unusual behavior. Further, in (b) - (d) the intermittency effect decreases considerably faster for larger time lags $\tau$ than in the wind field in Fig. 8. The analysis of the causes for these relevant and interesting differences would be an important outcome of the investigations. Here the authors merely refer to future studies. I would, however, expect the authors to make investigations into potential reasons and provide explanations in the present paper.

- Fig. 11: Also in the case of the unusual characteristics of the Kurtosis shown in Fig. 11 (strong drop for smaller time lags, strong peaks at $\tau$=1s, 2s,…), no analysis of the results and indication of possible causes was made. The statement "This might be directly related to the wind turbine and is subject to future research." is not sufficient here. Please check the BEM model and settings and ensure that no tower impact, tilt, gravity etc. is considered and elaborate the interpretation.

- P. 23: It would be interesting to compare the fatigue loads resulting from the original Mann model and the extended model and I suggest including the corresponding results in the paper to show the intermittency has a relevant impact on the loads.

- Conclusion: It would be nice if findings on the conversion of the intermittent wind field to the turbine loads and the fatigue loads could be added, see comment above.

- Figs. A1/A2: The thin grid lines and axis ticks are barely visible (in the printout). The lines in Fig. A2 should be thicker.

**Typos:**

- L- 152: Pope (2001) → (Pope (2001))

- L. 190: (s) / (t) → no brackets

- L. 250: space after „area."

- L. 270: „c=20" → „c=20s." + math mode in latex

- L. 321: "$\tau$ Fig. 9" → "$\tau$. Fig. 9"

- Tab. 4, caption: for consistency "Rel.Diff." → Rel.Diff"

- L. 369: "sec" → "seconds"

- L. 406: „…= 1" → „…= 1m"

---

## Author Comment (AC1)

**AC 1**

**"Applying a Random Time Mapping to Mann modelled turbulence for the generation of intermittent wind fields"**

Khaled Yassin, Arne Helms, Daniela Moreno, Hassan Kassem, Leo Höning, and Laura J. Lukassen:

We would like to thank the reviewer for the valuable comments, questions, and suggestions. They helped us significantly. We agree that the manuscript needs improvement and revised it as described below. All your comments have been addressed below in blue. Line numbers refer to the marked version of the manuscript "marked.pdf"

This manuscript presents a time-mapped Mann model to derive a synthetic wind field model. Wind field modeling plays an important role in wind energy studies. In general, it is an interesting topic.

1) The writing needs significant improvement. Please use scientific expressions in the manuscript.

Thank you for this comment, we have improved scientific writing. This comment has been applied in several locations of the manuscript and can be seen in the marked version.

2) There are some state-of-the-art field investigations on the influence of turbulence on the wind turbine structural response. Please discuss your results with the field measurement results.

A new paragraph has been added in lines 78-85 in the marked version of the manuscript where a new state-of-the-art paper was added and described in detail. Also, another source was discussed in lines 428-430

3) line 100: Please provide more information on how to choose the two points. Any restrictions about the distance between these two points.

Eq. 3 is a theoretical equation, which means in general that there are no restrictions on selecting the two points. However, when it is applied to our case, it is limited by the data set of points that we have from our fields. An additional statement has been added to the manuscript to clarify this issue below Eq. (3) in lines 116-118 in the marked version of the manuscript.

4) Fig.1: please add error bars on the measurement results since you have 500-time series samples for analysis.

All the 500-time series were combined and then the statistics of the combined time series are shown in this figure. This means that the statistics are not generated by the average over the number of individual time series. An additional statement in lines 158-159 in the marked version of the manuscript has been added to clarify this issue.

5) Sec 2.2 & Sec 2.3: please make it more concise.

We agree with the reviewer that these two sections contain several mathematical equations and descriptions of the underlying model. However, we find it beneficial to introduce the reader to the Mann and CTRW model since our new suggested model is a combination of the two. Furthermore, most of the described equations are used later in the analysis of the wind fields and for plotting the results of this analysis. An additional statement in lines 98-100 in the marked version of the manuscript has been added to clarify this issue. We have also omitted the scaling factor equation from lines 196-198 in the marked version of the manuscript.

6) Fig. 8: The curves are shifted vertically for visualization. Please provide more detailed information.

In principle, the curves should be on top of each other since they represent the probability of the velocity increment. However, we think that this would not be clearly represented. Therefore, each plot is shifted from the others for better visualization. Accordingly, the Y-axis of this plot is not the scale of these plots. The resulting PDFs for the increments have been similarly displayed in previous contributions on the topic (see Vindel et al. (2008), Liu et al. (2010), Mücke et al. (2011), Böttcher et al. (2007), Morales et al. (2011)). An additional statement in the caption of Figures 1 and 8 in the manuscript has been added to clarify this issue.

7) Why did you select 10% as the turbulence intensity level?

In the IEC 61400-1 standard, Turbine class B (average class) should be designed at TI = 0.14 that occurs at 15 m/s. By taking the same standard deviation value, a mean wind speed of 20 m/s corresponds to $TI_1 = 0.105$ which we have approximated to $TI_1 = 0.1$ (which in our paper is $TI_1$ since it is related to the wind speed component in streamwise $x_1$ direction). An additional statement in lines 286-287 in the marked version of the manuscript has been added to clarify this issue.

8) Why did you choose a fixed pitch and fixed speed wind turbine for the simulation? Most current utility-scale wind turbines are variable speed and variable pitch regulated.

As indicated in Sec. 1, the main scope of this paper is to introduce the generation of an intermittent wind field from the Gaussian Mann field using time mapping. This aims to investigate the statistics of the resulting field, and to investigate whether this intermittency is transferred to the loads. Here, it was important for us to work out the pure effect of intermittent wind on loads explicitly without additional effects such as pitch and variable speed. You are right that in principle this is an interesting question to be answered. A deeper investigation of the effects of intermittency on different loads of a

variable speed and variable pitch wind turbine and in other wind conditions is subject to future work. An additional statement in Sec. 5 in the marked version of the manuscript has been added to clarify this issue.

9) The mean wind speed in Table 3 is extremely high (20 m/s). In most wind farms, the annual average wind speed won't exceed 12 m/s. Please explain and add cases with lower speeds.

Thank you for this comment. You are right that 20 m/s is rather high. On the other hand, at 20 m/s the wind turbine is above rated wind speed without variable pitch and variable speed. The 20 m/s was an exemplary case as an extreme case just to show the effects of intermittency. However, other wind speeds are not in the scope of this paper and could be investigated in future work. Therefore, an additional statement in Sec. 5 in the manuscript has been added related to this issue.

10) Please add wind shear as well in the FAST simulations as well. It may have a larger impact on the turbine structural response than turbulence.

We agree that shear can be an important factor. In the Mann model, there exists the option to add shear which incorporates stretching of the turbulent eddies. Adding shear to our Time-mapped Mann model requires more consideration to be made. It is not as straightforward as in the Mann model, since the intrinsic time domain (which corresponds to one slice in the Mann model) is mapped to the physical time domain non-equidistantly. One option could be to first add the eddy stretching to the Mann model and then apply our time mapping. But this would certainly influence the stretching of the eddies in an unpredictable way. The consequences would have to be investigated in more detail, which exceeds the limits of the present work. But since we also consider this important, we have added an additional statement to the outlook in section 5.

11) Please add more statistical information about 500 sample datasets in the Appendix, including but not limited to wind direction, wind speed, turbulence level.

An additional description in lines 154-158 in the marked version of the manuscript has been added to clarify this issue.

---

## Author Comment (AC2)

AC 2

**"Applying a Random Time Mapping to Mann modelled turbulence for**

**the generation of intermittent wind fields"**

Khaled Yassin, Arne Helms, Daniela Moreno, Hassan Kassem, Leo Höning, and Laura

J. Lukassen:

We would like to thank the reviewer for the valuable comments, questions, and suggestions. They helped us significantly. We agree that the manuscript needs improvement and revised it as described below. All your comments have been addressed below in blue. Line numbers refer to the marked version of the manuscript "marked.pdf"

General comments:

The intermittency of atmospheric wind fields has already been examined in many studies in the past. The working group in Oldenburg has been working for a long time on the investigation of the influences of intermittent wind fields on the aerodynamic properties of airfoils and wind turbines and the development of methods for the synthetic generation of wind fields taking intermittency into account. The present manuscript represents a further contribution to this topic. It proposes the combination of the CTRW-based approach suggested by Kleinhans with the established Mann model for turbulence generation, in order to utilize the spatial correlations considered in the latter model. The differences to the original Mann model are exemplified by a generated wind field and the load statistics for a 1.5 MW turbine calculated with BEM by means of FAST. It could be shown that mean values and variances of the wind fields and loads agree well. However, the extended approach, denoted time-mapped Mann model, shows intermittencies, both, in the generated wind field and in the calculated loads except for the flapwise blade root bending moment. The consideration of shear, as foreseen in the Mann model, obviously requires further, considerable development effort in the CTRW approach and is still pending. The present results are therefore limited to a wind field without shear.

The manuscript is concisely written and contains a good and broad overview of the specific state of research. However, unconventional nomenclature, partial lack of introduction of variables in the text, and inconsistent use of variables complicate readability. The results show some peculiarities in the predicted loads that may be attributed to the turbine BEM model rather than to the wind field model.

You are right, there were some undefined nomenclature and other issues. We have modified the manuscript to make it as clear and readable as possible.

The turbine model should be checked prior to the interpretation of the results.

We have based our turbine simulations on a validate model.

In the results section, the authors largely limit themselves to describing the results visible in the pictures without analyzing and explaining peculiarities in the results. In this respect, the manuscript should be improved.

You are right, the explanation of some results was not clear. Therefore, we have elaborated the results to be clear as possible.

Moreover, I do not see why an Appendix is necessary and suggest incorporating the content directly in the paper.

We have added this appendix since the cubic case is only used to show some concepts related to the number of grid points. Therefore, we thought that writing this part as an appendix would make it easier for the reader to follow the mainstream of ideas.

In conclusion the study is relevant and I in principle support publication of the manuscript if the following comments are taken into account and the interpretation of the results is elaborated.

Thank you for your positive feedback. All comments are addressed in this document.

Specific comments and remarks:

- Please introduce new properties throughout the document (like e.g. r in L. 104, $\Delta x_2$ and $\Delta x_3$ in L. 115, ...) and use the variables consistently.

Noted and modified in both cases. Also, it is checked over the whole paper to be used consistently and can be seen in the marked version of the manuscript.

- L. 112: „where the double index does not indicate a summation here" is more confusing than helpful.

This sentence is now replaced in line124 of the marked version of the manuscript with "where the ii-index refers to the respective diagonal element of the tensor".

- L.132: Are $u_1$ and $u_2$ velocity fluctuations or absolute velocities? Compare eq. (1) small vs. capital letter.

You are right, $u_1$ and $u_2$ are the velocity fluctuations as indicated in Eq.1. However, in this line, we intend to show the absolute velocity of $U_1$ and $U_2$. The symbols are now corrected in line 152 of the marked version of the paper.

- L. 138f.: An increasing deviation for reduced $\tau$ is hardly visible in Fig. 1 due to the vertical shift. Perhaps this can be made visible by a more appropriate choice of the ranges shown.

You are right, the differences are hardly visible in Fig 1 (a). However, the differences are made visible through plotting the kurtosis in Fig 1 (b). In Fig 1 (b), a wider range of τ is studied to show the change of the kurtosis with τ. However, strong deviations from Gaussianity are observable even for large $\tau$ values due to the superposition of large-scale structures as explained by Böttcher et al. (2007). The work by Böttcher et al. has been accordingly mentioned and referenced in the paper in line 172. An additional explanation on this reason for the curves looking similar in Fig 1 (a) is given in lines 173-176.

- L172: "different" " multiple"/"several"

Noted and modified.

- Figure 2: To use x2 as vertical component and x3 as lateral is unusual (cf e.g., Mann,1998). Please ensure consistency throughout the paper. N as well as the Δ values likely refer to the Mann box parameters. These are not introduced yet, please do so. Using Δx 2 and Δx 3 with a different meaning than in equation (6) confuses. I suggest to use e.g., $\Delta x_{1,M}$ and $\Delta x_{2,M}$ here.

Thank you for this hint. This work has followed the same numbering as Mann (1988). However, in the previous version of the paper, the numbering was not correct in Fig.2. This mistake is now fixed, and the figures are updated.

For $\Delta x_1$, $\Delta x_2$, and $\Delta x_3$ describing the Mann box, they are now called $\Delta x_{1,m}$, $\Delta x_{2,m}$ and $\Delta x_{3,m}$ as suggested. This has been modified in Tab. 1 and appendix A1

- L. 255ff: Is there a specific reason why you prefer Eq. (21) instead the derivation according to IEC 61400-01 if no shear is considered? L=0.8*Λ with Λ = 42m due to z hub > 60m →L=33.6m

Yes, you are right that the IEC61400-1 gives another method to calculate the length scale. However, according to *Hannesdóttir et al. (2019)*, the turbulence length scale was calculated using Eq. 21 (now Eq. 20 in the updated version of the manuscript) because the turbine is large, and the turbulence length scale should be of the same order of magnitude as the hub height. We have used this equation to also use it for even larger wind turbines. An additional statement has been added to the manuscript to clarify this issue in lines 293-295 in the marked version of the manuscript.

- Table 1: Please justify the choice of the Mann box parameter. How is length n x1 chosen? n*Δx in lateral direction is not even twice the integral length scale L; is this sufficient to resolve the statistics?

In general, the parameters of the Mann box were selected to be a compromise between the reasonable representation of the wind field and the computational resources needed for the simulations. In Fig. 6 we have provided a comparison between spectra of the Mann field, the time-mapped Mann field, and the theoretical spectra calculated from Eq. 14, where the plots showed good agreement. Hence, we assume that the correlation

can be displayed sufficiently using this length scale value and that the current statistics are sufficient. At the same time, the computational effort is still reasonable.

$n_{x1}$ was chosen to achieve a simulation time long enough to perform the appropriate field analysis and to be within the limits of the number of points in the Mann box generation tool. So, $\Delta x_{1,m}$ was calculated according to $U_1$ and $n_{x1}$.

Also, the values of $\Delta x_{2,m}$ and $\Delta x_{3,m}$ are set to be close to $\Delta x_{1,m}$ to avoid any problems that could happen due to high aspect ratios of the grid cells. Accordingly, $n_{x2}$ and $n_{x3}$ are selected to cover the whole turbine without exceeding the numerical limits of the tool and to keep the field big enough to perform different statistical analyses in $x_2$ and $x_3$ directions. We use $n_{x2}=n_{x3}$ and $\Delta x_{2,m}=\Delta x_{3,m}$ because of the symmetric shape of the turbine and because there are no expected differences in the field in $x_2$ and $x_3$ directions. Accordingly, it is reasonable to make the parameters of the Mann box in $x_2$- and $x_3$- directions the same. The value for the Mann box parameter $c_K \varepsilon^{2/3}$ was introduced in the ESDU spectral model.

- L. 269ff: As Kleinhans pointed out, one disadvantage of the CTRW approach is the definition of the various parameters (α, c, $\Delta s_t$ ) that control the intermittency. The general validity of the approach is limited in particular by the fact that $\alpha$ depends on the time lag and must be calibrated in order to correctly represent real intermittent wind fields. The values chosen in the present study do not contain a $\alpha(\tau)$ dependency and differ significantly for c, Δs t but also for α from the values chosen by Kleinhans. Are there any recommendations for selecting these parameters for practical cases and can the authors reason their choice?

The selection of the parameters was based on the calculated kurtosis (from Eq. 11) of $v_1$ for the generated time-mapped signals. An iterative process was performed for achieving the target values of Kurt($v_1$). These target values are in a comparable range to the Nordex data in Fig. 1 and to those presented in *Schwarz et al.* (2020) for different atmospheric measurement data sets. The main objective of the iteration was to achieve intermittent behavior without compromising the other statistical features such as the standard deviation and the spectrum. In general, as already mentioned in Sec. 2.3, lower values of α and higher values of c and $\Delta s_t$ will lead to more intermittent time series. An additional statement has been added to the manuscript to clarify this issue in lines 308-309 in the marked version of the manuscript.

- L. 279: Repetition of the statement from L. 276f.

The sentence in line 276 is now deleted.

- L. 283: Please include your definition of ($\kappa$ =2*π*f/U) and specify U to make sure whether it's an angular wavenumber.

The definition has been added to the manuscript to clarify this issue in line 319 in the marked version of the manuscript.

Since in this section we apply Fourier transform in space and this results in the wavenumber in space, we do not want to confuse it with the frequency space. If the comment is referring to the Ref. Mann (1998), where f is either the frequency or the Coriolis parameter in Equ. 18 of this reference, then we do not have the second term of Equ. 18 i.e. f is the frequency as described in Ref. Mann (1998).

- Fig. 6: Fig. (a) obviously shows a narrowband spectrum whereas (b) and (c) seem to be smoothed or have large frequency bins. Please justify this inconsistency or align.

Fig. 6(a) represents the spectrum in $x_1$-direction with $nx_1 = 131072$ while Figures 6(b) and (c) represent the spectra in $x_2$- and $x_3$-directions with $nx_2 = nx_3 = 32$. The lower number of grid points in $x_2$- and $x_3$-directions caused these smooth curves in Figures 6(b) and (c) after using the Fourier transform to generate the spectra. An explanation statement has been added to the manuscript to clarify this issue in lines 366-368 in the marked version of the manuscript.

- L 291ff: Please rework interpretation of Fig 6 and explain what spectra characteristic was to be expected for the new wind field model (e.g. L. 293: Does the difference between Time-mapped and original Mann in 6(a) for low wavenumber represent intermittency?

That is a good remark. It is important to note that the difference between the Time-mapped and original Mann field in Fig. 6 does not represent the intermittency. As indicated in Sec. 2.1, the velocity spectrum is not sufficient to show the intermittency. Therefore, increment statistics should be used in this case to show the effects of intermittency. Here, an explanation statement has been added to the manuscript to clarify this issue in lines 358-359 in the marked version of the manuscript.

- L. 294f: The statement cannot be followed because the small wavenumber range is not shown in plots 6(b) and (c).

You are right, we cannot plot the spectra for low wavenumbers because of the small number of grid points in $x_2$- and $x_3$-directions. However, the deviation happens for high wavenumbers. We have rephrased this in lines 356-358 in the marked version of the manuscript.

- L296: The number of grid points in vertical and lateral direction is the same for the Mann model and the time-mapped Mann model, Therefore I cannot follow your explanation that the differences result from the grid. Please clarify.

The Time-mapped Mann field is generated from time mapping of the different slices of the original Mann field. To get the required time step in the time-mapping process, the slices of the time-mapped field must be regenerated using linear interpolation. This linear interpolation between the time-mapped slices and the regenerated slices leads to interpolation errors and hence the deviations, as shown in Fig.3. and lines 351-353

- L. 297ff: The conclusion from comparison to A1 is not obvious. Ranges are different and therefore deviations are hardly comparable. Moreover, changes in grid size and resolution are mixed and hence effects cannot be attributed to a finer discretization for sure. It is still unclear why the size or resolution should explain differences between time-mapped and original Mann model.

For each box, there are three different parameters used to generate the box: the overall box size, the grid cell size, and the number of cells. It is not possible to change the value of one of these parameters alone without changing at least one of the other two parameters to generate a grid with comparable results. Therefore, we have generated a new grid with cubic cells to have independence of the direction since all directions have the same grid points and cell sizes. This explanation has been added to the appendix of the manuscript in lines 479-483.

- L. 308ff: Please mention the "settings" ($\Delta x_2$ , $\Delta x_3$) also in the text and ensure that this distance values are not confused with Mann box parameters and elaborate the interpretation.

Noted and modified in the manuscript to clarify this issue in lines 291-292 in the marked version of the manuscript. We have also renamed these parameters to $\Delta x_{1,m}$ $\Delta x_{2,m}$, and $\Delta x_{3,m}$ not to be confused with the spatial steps in the statistical calculations.

In the comparison between Mann and Time-mapped Mann fields, the velocity spectra are expected to be the same since the one-point spectra are not able to show the effects of the time mapping. Therefore, two-points statistics, like coherence, are used to show these effects. In Fig. 7, we can notice the differences between Mann and Time-mapped Mann fields in the $x_1$-direction while $x_2$- and $x_3$-directions are almost similar. The same can be noticed in Fig. A2. This explanation has been added in lines 356-357 in the marked version of the manuscript.

- L. 311ff: Can the authors please interpret this characteristic in more detail and explain the difference to the behavior for the cubic case in Appendix? The plots in A2 are somehow smoothed, please get it consistent with Fig. 7.

Both, Fig. A2 and Fig. 7 were plotted in the same way. The noise that can be seen in Fig. 7 comes from a larger number of points in the $x_1$-direction in the original field. However, the lower number of points of the cubic case compared to the original case makes plot A2 smoother. An additional statement has been added to the manuscript to clarify this issue in lines 366-368 in the marked version of the manuscript.

- Fig 7: Please do not mix (x,y,z) and ($x_1$ ,$x_2$ ,$x_3$) and get it consistent with Fig. 2.

Noted and applied over the whole text.

- P. 19, Tab. 3: I assume that the Dynamic Stall and the Dynamic Inflow models were activated in the FAST calculations. To be on the safe side, this should be mentioned. Moreover, does the turbine have tilt?

Noted and modified in Tab. 3. Also, the turbine has a 5° tilt.

- Fig. 10: For better differentiation of the figures, the specific moment (RootFlap, …) depicted in (a) to (d) could be added to the axis labels or the respective figures.

Noted and the plots were modified as requested.

- L. 357ff: Fig. 10 shows interesting results. Fig. 10 (b) - (d) clearly shows the influence of intermittency. However, Fig. 10 (a) does not show any impact of intermittency in the results with the new model. This is strange since I cannot see how the blade bending loads and the thrust (as the sum of the three blade loads in the same direction) differ in their intermittent behavior that strongly. Please conduct further investigations to explain this unusual behavior. Further, in (b) - (d) the intermittent effect decreases considerably faster for larger time lags $\tau$ than in the wind field in Fig. 8. The analysis of the causes for these relevant and interesting differences would be an important outcome of the investigations. Here the authors merely refer to future studies. I would, however, expect the authors to make investigations into potential reasons and provide explanations in the present paper.

The RootFflap in Fig 10 shows a significant difference to the other loads. In fact, the RootFlap increment PDF seems to be non-intermittent at all. However, that is not entirely correct, it just turns out that it exhibits oscillations like Fig 11 but with much lower amplitude and on a different frequency which is not captured by the specific selected time step sizes in Fig 10 a.

Conclusions about the faster decrease of the intermittency for loads compared to wind need further investigation. The investigation of how different spatial correlations of the wind (e.g non-correlated, fully correlated) affect the intermittency of the loads might explain this difference. However, in *Mücke et al.* (2011), the authors have shown that the kurtosis of the measured wind is higher than the kurtosis of the resulting turbine torque (the torque was calculated from numerical simulations by using atmospheric wind measurements as input). These ideas were elaborated in lines 428-430.

- Fig. 11: Also, in the case of the unusual characteristics of the Kurtosis shown in Fig. 11 (strong drop for smaller time lags, strong peaks at $\tau$ =1s, 2s,...), no analysis of the results and indication of possible causes was made. The statement "This might be directly related to the wind turbine and is subject to future research." is not sufficient here. Please check the BEM model and settings and ensure that no tower impact, tilt, gravity etc. is considered and elaborate the interpretation.

We have double-checked the settings of our BEM model and we can confirm that the predicted response of the turbine is due to the characteristics of the wind field, and not due to the turbine model. Further analysis of the results has shown that the effect of the intermittency is related to the rotational frequency of the turbine (see line 433 in the marked version of the manuscript).

- P. 23: It would be interesting to compare the fatigue loads resulting from the original Mann model and the extended model and I suggest including the corresponding results in the paper to show the intermittency has a relevant impact on the loads.

You are right, it would be interesting to compare the resulting fatigue loads. However, it is important to first investigate whereas the proposed guidelines for calculating fatigue loads can properly detect the intermittent structures. The proposed rainflow counting method focuses on the amplitude of the oscillations rather than their temporal scales, that correspond to a key aspect of the intermittency. Given the discrepancies between the current definition of fatigue loads and the nature of the intermittency, it is questionable how accurate they can be correlated. In *Mücke et al* (2011), the authors concluded that the resulting larger fluctuations on the torque from an intermittent wind field were not detected by the rainflow cycle counting algorithm.

- Conclusion: It would be nice if findings on the conversion of the intermittent wind field to the turbine loads and the fatigue loads could be added, see comment above.

Noted and added to the conclusion.

- Figs. A1/A2: The thin grid lines and axis ticks are barely visible (in the printout). The lines in Fig. A2 should be thicker.

Plots are modified as requested.

Typos:

- L- 152: Pope (2001) →(Pope (2001))

Noted and corrected in line 183 of the marked version of the manuscript.

- L. 190: (s) / (t) →no brackets

Noted and corrected in line 221 of the marked version of the manuscript.

- L. 250: space after „area.“

Noted and corrected in line 282 of the marked version of the manuscript.

- L. 270: „c=20“ →„c=20s.“ + math mode in latex

The parameter c is dimensionless, refer to Equ. 19 and line ($0 < \tau_\alpha < c$) where $\tau_\alpha$ is dimensionless. Math mode is now used and corrected in line 305 of the marked version of the manuscript.

- L. 321: "$\tau$ Fig. 9" →"$\tau$. Fig. 9"

Noted and corrected in line 305 of the marked version of the manuscript.

- Tab. 4, caption: for consistency "Rel.Diff." →Rel.Diff"

Noted and corrected in Tab. 4 of the marked version of the manuscript.

- L. 369: "sec" →"seconds"

Noted and corrected in line 417 of the marked version of the manuscript.

- L. 406: „...= 1" →„...= 1m"

Noted and corrected in line 459 of the marked version of the manuscript.

---

## Referee Report (RR1)

Review of the revised manuscript:

*Khaled Yassin, Arne Helms, Daniela Moreno, Hassan Kassem, Leo Höning, and Laura J. Lukassen:*
**"Applying a Random Time Mapping to Mann modelled turbulence for the generation of intermittent wind fields"**

**General comments:**

Thanks to the authors for answering most of my questions in the response and considering my suggestions in the revised version of the manuscript!

However, the authors did not provide a sufficient explanation for the conspicuous behavior of the root bending loads (Fig. 10a), where, in contrast to the other evaluated loads, no heavy tails can be observed. The authors mention that fluctuations do occur also in the RootFlap loads, but with smaller amplitude and a different frequency, which is not resolved with the selected specific step size. What time step size would be required for this and why was the step size not adjusted in the results presented to visualize the effect? How big are the differences in the amplitudes and frequencies of the different loads (quantitatively)? The authors have now added the important information in table 3 that a tilt angle is considered in the turbine model, which leads to deterministic 1P or 3P load fluctuations. Whether gravity forces were considered remains open. The authors mention 3P loads in the reply to the reviewer, but not in the revised manuscript. I would like to ask the authors to explicitly and completely mention the modeled turbine characteristics that lead to deterministic force fluctuations and to add the influence of deterministic load variations in the discussion and interpretation of Figs. 10 and 11. I would likewise ask the authors to better explain the strikingly different characteristics of the RootFlap loads compared to the other loads. Perhaps time series of the calculated four loads can help here. I would have expected that intermittency of the wind would be transferred to the loads, but this inconsistent behavior of the RootFlap loads requires an explanation if the turbine loads are to be included in the manuscript. The development work on the combination of the CTRW with the Mann model is independently relevant and worth to be published.

**Minor remarks:**

Below are some minor comments that should be considered in the final version, with line numbers referring to the revised manuscript.

- L. 309: If the iterative procedure, as described in the reply to the reviewer, is not explained in the cited publication by Schwarz, I suggest to add a sentence on the procedure in the manuscript.

- L. 319: I can't see an addition of the definition of $\kappa$ and U here.

- L. 339/340: In your reply to my last review, you attributed the differences between the Mann and Time-mapped Mann models in Fig. 6(b) and (c) to the interpolation in the x-direction. That may be. However, I still cannot understand the second part of the sentence "...and in this case due to the low number of grid points in the transverse directions".

---

## Referee Report (RR2)

**Review of "Applying a Random Time Mapping to Mann modelled turbulence for the generation of intermitten wind field" by K. Yassin *et al* submitted to Wind Energy Science**

Jakob Mann

August 2022

**General remarks**

The authors propose and investigate a modification to a three-dimensional turbulence inflow model, the so-called Mann model. The modification the authors propose is motivated by an analysis that shows that velocity increments (differences over relatively short time scales) appear not to be Gaussian as it is implied in the Mann model. This is called intermittency. The way to authors do the modification is to randomly stretch and compress the time axis without changing the overall or average progression of time. This has been done previously for a one-dimensional field but never, to my knowledge, to a three-dimensional field. The authors finally test the impact of their modified fields (Time-mapped Mann fields) relative to unmodified fields on wind turbine loads. They show the standard deviations of four different loads are essentially unchanged while the increment statistics differs. The kurtosis of the increments of three out of four loads are increased albeit not as much as the velocity field itself.

Although the research as such is original, the are several severe issues with the paper. Let me summarize those in the following points:

1. The motivation for intermittency (figure 1) is misleading.

2. The modification to the Mann model is quite nonphysical.

3. The conclusion that the spatial structure of the turbulence (understood as the second-order statistics) is unchanged by the modification is flawed.

4. Some of the simulation results are hard to understand and are not well explained.

**Point 1**

I might misunderstand the background for figure 1, so I'll summarize it here. The plot is compiled from data where the 10-minute average wind speeds ranges from 5 to 15 m/s and the turbulence intensity from 5% to 25%. Imaging that you only have one wind speed but that the turbulence intensity is 5% at night and 25% during the day, and during these periods the turbulence is perfectly Gaussian. Now you calculate the increment pdf which for both night and day are Gaussian with the night pdf being much narrower than the day pdf. But when you add them, they total pdf will become non-Gaussian with a positive kurtosis excess. The situation become more complicated with you included varying mean wind speed, but the example illustrates that you can get small-scale intermittency simply by combining Gaussian distributions with different widths. This is surely not what we are after. I wonder if one takes a long, stationary chunck of the data and do the same analysis whether you get strong kurtosis at all.

Another more puzzling point is the equation for the increment

$$v_{meas} = \sqrt{u_1^2(\boldsymbol{x}, t+\tau) + u_2^2(\boldsymbol{x}, t+\tau)} - \sqrt{u_1^2(\boldsymbol{x}, t) + u_2^2(\boldsymbol{x}, t+\tau)} \qquad (1)$$

appearing in the text on page 6. So according to the definition the authors take the length of the fluctuating part of the horizontal vector and subtract that at times separated by $\tau$. This is a very strange procedure. Once could understand if one took the length of the total vector in which case the square root would be roughly equal to $U_1 + u_1$ (see L. Kristensen, J. Atm. Oc. Tech. 1998). As it stands now, any perfectly joint Gaussian $u_1$ and $u_2$ process would give a kurtosis excess of $v_{meas}$.

I think the motivation section should be improved answering these critical questions.

**Point 2**

The mathematics of the modification of the Mann model is quite understandable. However, it is not very physical. For example, why it the focus only on the kurtosis? The skewness remains zero in the modified field although this is the property that is known to be non-zero according to Kolmogorov 1941. The method modifies the intermittency in the $x_1$ direction but it remains perfectly Gaussian in the transversal direction, so the procedure introduces a small-scale anisotropy that there is no experimental evidence for. The resulting field becomes compressible (maybe it doesn't matter to much for loads, but it is a bit unphysical). The authors are only using isotropic turbulence which is far from what is observed in the atmosphere where $\Gamma$ is usually between 3 and 4. The explanation for this on page 26 is not convincing. Is it difficult to generate a $\Gamma \neq 0$ field and then apply the time mapping? A relatively simple and more physical method to generate fields with non-Gaussian increment was presented by Rosales and Meneveau, Phys. Fluids 2006 which I think should be discussed

as well. The method of Berg et al (2016) is also more physical and they see small effect on the damage equivalent loads. Please discuss what is physical and what is not.

**Point 3**

Coherence (figure 7) is a second-order statistics and it certainly changes a lot. (Please don't show all the irrelevant scatter in the plots, just the smoothened coherences.) I don't think this large change in coherence towards a much more pointed shape has been observed anywhere in atmospheric measurements while the theoretical von Karman coherences have been verified for small separations at several occasions. Please comment on this. Since the two-point cross spectra changes so drastically you should also expect the auto-spectra to change (figure 6). This is obscured in figure 6 in the way the spectra are treated and plotted. First of all, it is costumary, and with good reason, to plot the pre-multiplied spectrum ($\kappa_1 F_1(\kappa_1)$) because it makes it easier to see how the variance is distributed on frequency. Secondly, please do bin averaging so you plot an equal number of power spectral densities per decade. In this way it is possible to see the differences between the conventional and time-mapped spectra. Also regarding figure 6, I think it is totally unphysical to assume uncorrelated time mapping at every point and it obviously give nonsensical $F_2(\kappa_2)$ and $F_3(\kappa_3)$ spectrum. There are no reason to show these.

**Point 4**

There seems to be no good explanation on the very non-intermittent behavior of root-flap moment in figure 10. It is also hard to understand the behavior of the kurtosis in figure 11. Why do you see very regular peaks at $\tau$ equal to interger seconds?

**Some specific remarks**

1. In the introduction, which is nice, it would be great to be more specific on what the different studies show. Please state how large are the difference in percent instead of stating "very close", "agree quite well", "are different from" wherever it is possible.

2. Is there any physical reason for the choice the distribution of time increment maybe related to the fact that it is $\alpha$-stable? Eq (17) seems to miss "p()" on the left hand side.

3. Smaller language issues. l 37 "realistic" → "realistically", l 110, I think it is more correct to use "componennt" instead of "direction", l 112 "statistics are" → "statistics is" (also l 319), l 115 "independent from" → "independent of", l 127 "*coh*" should be "coh", l 225 A distribution is not delta correlated but it can have a delta destribution.

4. Eq (13) A square is missing on the $\kappa$ after the Kronecker delta symbol.

**My conclusion**

This paper contains new and original research. However, many changes are needed in order to get it up to the required scientific standard.

---

## Referee Report (RR3)

Review of the 2nd revision of the manuscript:

*Khaled Yassin, Arne Helms, Daniela Moreno, Hassan Kassem, Leo Höning, and Laura J. Lukassen:*
**"Applying a Random Time Mapping to Mann modelled turbulence for the generation of intermittent wind fields"**

I would like to thank the authors for the answers to my remarks and questions to the revised version of the manuscript, the addition of Appendix B and the modification of Fig. 10, in which the time lags considered have now been selected in such a way that heavy tails are also visible for the RootFlap moments! The authors have completed Table 3 with the information on the turbine model and it is now clear that a tilt angle and gravity effects have been considered in the model, each of which produces deterministic 1P load variations for each blade. However, this supplementary information and the updated result in Fig. 10 raises a new fundamental question for me. The adaptation of the considered time lags in Fig. 10a has led to a sudden appearance of heavy tails in the root-flap moments that were not visible in the results of the first two versions of the manuscript. In my opinion this means that the heavy tails now visible in Fig. 10a are obviously caused by the deterministic 1P loads and not by the intermittency of the wind field. Fig. B1 in Appendix confirms the prominent 1P peak as well as the higher harmonics in the spectrum of the root-flap moments. A further indication that the heavy tails visible in Fig. 10a may be caused by deterministic loads results from the fact that the heavy tails strongly decay for higher time lags, while they are still clearly visible for the wind field (Fig. 8). A comparable behavior of decaying heavy tails for larger time lags can found in Fig. 10 for the other loads, so that also here it can be suspected that the visible heavy tails could be caused by deterministic 3P loads and not by the wind field.

If the heavy tails visible in the results for the time-mapping Mann model are caused by the deterministic loads, the question arises why the calculations with the original Mann model do not show heavy tails at all for any time lag (Fig. 10). Actually, I do not understand these results and would like to ask the authors to carefully check the calculations and the setups and provide more explanations. Are the turbine models really consistent? For the time being and with the available information and results, I cannot agree tostro the main conclusion of the manuscript (line 479ff), according to which the intermittency of the wind field generated by means of the time-mapped Mann model is transferred to the rotor loads. In order to demonstrate the influence of the intermittency of the wind field and to justify this conclusion, I strongly suggest to first consider a rotor without tilt and without gravity forces, i.e. without deterministic load variations, for both wind fields and I would be happy if my open questions and concerns could be solved.

---

## Author Response (AR2)

**Author's Responses 1**

Thank you for the valuable feedback on our manuscript. We addressed the remarks in the comments below.

Please note that reviewers comments are written in green, and the authors' responses are written in black.

Please also note that all line numbers in the responses refer to the line numbers in the marked version of the manuscript.

**General comments:**

Thanks to the authors for answering most of my questions in the response and considering my suggestions in the revised version of the manuscript! However, the authors did not provide a sufficient explanation for the conspicuous behavior of the root bending loads (Fig. 10a), where, in contrast to the other evaluated loads, no heavy tails can be observed. The authors mention that fluctuations do occur also in the RootFlap loads, but with smaller amplitude and a different frequency, which is not re- solved with the selected specific step size. What time step size would be required for this and why was the step size not adjusted in the results presented to visualize the effect? How big are the differences in the amplitudes and frequencies of the different loads (quantitatively)?

Thank you for this remark. The energy spectra of the thrust, torque, and tower fore-aft bending moment show peaks at the 3P frequencies and higher whereas the blade root flapwise bending moment also shows a peak for the 1P frequency. This carries over to the kurtosis spectrum shown in figure 11, which shows also the 3P frequency for the thrust. For the blade root flapwise bending moment, the frequency of fluctuations relates to the 1P frequency of the load energy spectrum. Therefore, the time step was changed in Fig. 10(a) to make the intermittency of the blade root flapwise bending moment visible in the plots. This was clarified more in lines 441-447. Also, Appendix B was added for more explanation.

The authors have now added the important information in table 3 that a tilt angle is considered in the turbine model, which leads to deterministic 1P or 3P load fluctuations. Whether gravity forces were considered remains open.

For the BEM simulations the gravity was enabled but the blades were considered rigid. These two points are now added to Turbine's characteristics in Table 3.

The authors mention 3P loads in the reply to the reviewer, but not in the revised manuscript. I would like to ask the authors to explicitly and completely mention the modeled turbine characteristics that lead to deterministic force fluctuations and to add the influence of deterministic load variations in the discussion and interpretation of Figs. 10 and 11. I would likewise ask the authors to better explain the strikingly different characteristics of the RootFlap loads compared to the other loads. Perhaps the time series of the calculated four loads can help here. I would have expected that the intermittency of the wind would be transferred to the loads, but this inconsistent behavior of the RootFlap loads requires an explanation if the turbine loads are to be included in the manuscript. The development work on the combination of the CTRW with the Mann model is independently relevant and worth to be published.

Thank you for this remark. The energy spectra of the thrust, torque, and tower fore-aft bending moment show peaks at the 3P frequencies and higher whereas the blade root flapwise bending moment also shows a peak for the 1P frequency. This carries over to the kurtosis spectrum shown in figure 11, which shows also the 3P frequency for the thrust. For the blade root flapwise bending moment, the frequency of fluctuations relates to the 1P frequency of the load energy spectrum.

In order to clarify this more in the manuscript, we have added plots for the energy spectra of the blade root flapwise moment and the thrust in the appendix. Further, we have adapted the choice of the time step for the plotting in figure 10a in order to show the intermittency more clearly. To clarify this, we have added a text in line 441-447. Also, Appendix B was added for more explanation.

**Minor remarks**

**1.)** L. 309: If the iterative procedure, as described in the reply to the reviewer, is not explained in the cited publication by Schwarz, I suggest to add a sentence on the procedure in the manuscript.

Thank you for the suggestion. We added a sentence in lines 315-318.

**2.)** L. 319: I can't see an addition of the definition of $\kappa$ and $U$ here.

The definition of $\kappa$ and $U$ is now added to line 329.

**3.)** L. 339/340: In your reply to my last review, you attributed the differences between the Mann and Time-mapped Mann models in Fig. 6(b) and (c) to the interpolation in the x-direction. That may be. However, I still cannot understand the second part of the sentence "...and in this case due to the low number of grid points in the transverse directions".

You are right, this sentence needs more clarification. We mean that the deviations between Mann and Time-mapped Mann fields happen due to interpolation errors that are more obvious in the transverse directions due to the low number of grid points, and accordingly low number of interpolation points, in the transverse directions. This clarification has been added to the manuscript in lines 348-351.

**Author's Responses 2**

Thank you for the valuable feedback on our manuscript. We addressed the remarks in the comments below.

Please note that reviewers comments are written in green, and the authors' responses are written in black.

Please also note that all line numbers in the responses refer to the line numbers in the marked version of the manuscript.

**General remarks**

**Point 1:**

I might misunderstand the background for figure 1, so I'll summarize it here. The plot is compiled from data where the 10-minute average wind speeds ranges from 5 to 15 m/s and the turbulence intensity from 5% to 25%. Imaging that you only have one wind speed but that the turbulence intensity is 5% at night and 25% during the day, and during these periods the turbulence is perfectly Gaussian. Now you calculate the increment pdf which for both night and day are Gaussian with the night pdf being much narrower than the day pdf. But when you add them, they total pdf will become non-Gaussian with a positive kurtosis excess. The situation become more complicated with you included varying mean wind speed, but the example illustrates that you can get small- scale intermittency simply by combining Gaussian distributions with different widths. This is surely not what we are after. I wonder if one takes a long, stationary chunck of the data and do the same analysis whether you get strong kurtosis at all.

Thank you for this observation. In fact, 10-minute time series with such significant statistical differences, especially in terms of the standard deviation, might lead to a miscalculation of the kurtosis. In order to avoid this error, we now constrain the analyzed data to have comparable statistical characteristics of the wind velocity. Therefore, we selected a subset of time series whose values of mean wind speed and turbulence intensity are contained within a specific range. The range is defined as $9 \pm 1 \text{m/s}$ for the mean wind speed and $0.15 \pm 0.02$ for the turbulence intensity. Further, we also constrain the data in terms of their wind direction. For that, we consider the mean direction calculated over the 10-minute period. Then, we define a range of $\pm 20°$ from the main wind direction at the specific site. After filtering the data by mean wind speed, turbulence intensity and mean wind direction, 20 out of the 497 available time series are investigated.

Figure 1 has been modified accordingly. As can be observed, the values of the kurtosis significantly drop when considering the constrained measured data. In agreement with your comment, combining time series with highly different mean and standard deviation might result in values of kurtosis mainly driven by the superposition of Gaussian distributions with different widths, rather than by the phenomenon of intermittency in the wind. Nevertheless, values of kurtosis higher than 3 are still obtained after the filtering process of the measured data. That means intermittent features are identified in the statistics of the velocity increments which do not originate from the superposition of Gaussian distributions with different standard deviations. Consequently, figure 1 shows intermittent characteristics observed in the atmospheric wind, which we aim to reproduce in our work.

To clarify these points, lines 149-163 were modified in the marked version of the manuscript.

Another more puzzling point is the equation for the increment

$$v_{meas} = \sqrt{u_1^2(\boldsymbol{x}, t + \tau) + u_2^2(\boldsymbol{x}, t + \tau)} - \sqrt{u_1^2(\boldsymbol{x}, t) + u_2^2(\boldsymbol{x}, t + \tau)}$$

appearing in the text on page 6. So according to the definition, the authors take the length of the fluctuating part of the horizontal vector and subtract that at times separated by $\tau$. This is a very strange procedure. Once could understand if one took the length of the total vector in which case the square root would be roughly equal to $U_1 + u_1$ (see L. Kristensen, J. Atm. Oc. Tech. 1998). As it stands now, any perfectly joint Gaussian u1 and u2 process would give a kurtosis excess of $v_{meas}$. I think the motivation section should be improved answering these critical questions.

The definition of the increments $v_{meas}$ of the measured wind velocity was not correctly formulated. For an accurate formulation, we consider the definitions introduced for our previous reply in this document. Then, we define
$v_{meas} = u_{meas}(\text{x}, \ t + \tau) - u_{meas}(\text{x}, t)$. This formula is now in line 164 in the marked version of the manuscript. Also, line 166 and 177-184 in the marked version were modified to reflect the correct idea behind this analysis.

**Point 2:**
The mathematics of the modification of the Mann model is quite understandable. However, it is not very physical. For example, why it the focus only on the kurtosis? The skewness remains zero in the modified field although this is the property that is known to be non-zero according to Kolmogorov 1941. The method modifies the intermittency in the x1 direction but it remains perfectly Gaussian in the transversal direction, so the procedure introduces a small-scale anisotropy that there is no experimental evidence for. The resulting field becomes compressible (maybe it doesn't matter to much for loads, but it is a bit unphysical).

It is correct, that the combination of the Mann model and the time-mapping from the CTRW model does not yield all the desired statistics and that there is only temporal intermittency introduced. However, we wanted to go one step further from the idea of the CTRW model as used e.g. in (Schwarz: Wind turbine load dynamics in the context of intermittent atmospheric turbulence, Ph.D. thesis, Oldenburg, 2020) where the different time series are spatially uncorrelated - or fully correlated when the same time series is used many times in a certain area.

An important factor is to compare Gaussian and non-Gaussian wind fields and their effect on loads. Ehrich (Ehrich: Analysis of the effect of intermittent wind on wind turbines by means of CFD, Ph.D. thesis, Oldenburg, 2022) worked with CTRW wind fields which he time-shifted in order to create intermittent wind fields and compared to their Gaussian counterparts. He found an effect of intermittency on turbine loads.
Berg et al. (Berg et al.: Gaussian vs non-Gaussian turbulence: impact on wind turbine loads, Wind Energy 19, 2016) used LES simulations which naturally show non-Gaussian characteristics as input to wind turbine load calculations and performed POD in order to create a Gaussian counterpart from that. It is correct, that their outcome was that the effect of intermittency on loads is not very pronounced. However, we know from literature, that there is still a lot unknown concerning the effect of intermittency on loads, since different studies come to different conclusions. Our intention was to create a non-Gaussian wind field which shares commonalities with a Gaussian wind field in order to isolate the effect of intermittency based on a synthetic wind field model such as the Mann model.

From general turbulence theory, there are certain attempts to generate fields with non-Gaussian

increment statistics from a proper superposition of Gaussian fields.

Rosales and Meneveau (A minimal multiscale Lagrangian map approach to synthesize non-Gaussian turbulent vector fields, Physics of Fluids 18, 2006) perform a superposition of Gaussian Fourier modes and rescaled in order to receive a desired spectrum. This points to the superposition of Gaussian pdfs with varying variances as done by Castaing (Castaing et al.: Velocity probability density functions of high Reynolds number turbulence, Physica D 46, 1990), which has also been applied by Wilczek (Wilczek: Non-Gaussianity and intermittency in an ensemble of Gaussian fields, New Journal of Physics 18, 2016) who did superpositions of entire Gaussian fields. A similar approach has also recently been applied to the Mann model by a group from Oldenburg where the small-scale Gaussian statistics of Mann wind fields with varying covariances were superposed to yield small-scale intermittency (Friedrich, J., et al. "Superstatistical Wind Fields from Pointwise Atmospheric Turbulence Measurements." PRX Energy, vol. 1, no. 2, 2022, https://doi.org/10.1103/prxenergy.1.023006. )

However, our intention was to combine well established models in wind energy which we did by combining the Mann model with the time-mapping procedure from the CTRW model. It is always a matter of what is investigated, our model shows spatial correlations in the $x_2$-$x_3$-plane similar to the Mann model and allows to investigate the influence of temporal intermittency. to address your comment, lines 356-367 of the marked version of the manuscript were modified.

The authors are only using isotropic turbulence which is far from what is observed in the atmosphere where $\Gamma$ is usually between 3 and 4. The explanation for this on page 26 is not convincing. Is it difficult to generate a $\Gamma \neq 0$ field and then apply the time mapping?

Considering the $\Gamma$-parameter, we wanted to investigate the isolated effect of the time-mapping procedure without additional effects which influence the spatial correlations. This way, all discrepancies certainly come through the time-mapping procedure and the corresponding necessary interpolation procedure. Of course, a $\Gamma \neq 0$ Mann field could be time-mapped. However, there remain open questions. Of course, the shear rate is equal over the whole domain, but the mean velocity is changing. An open question is if the time-mapping should be equal over the whole $x_2$-$x_3$-plane or if it should be enhanced in areas with a supposed larger mean velocity. Such considerations are definitely of high value, however, since we aimed at investigating the general characteristics of the Time-mapped Mann field, this would have exceeded the scope of the paper.

We clarified more on that starting in line 486-493 in the marked version of the manuscript.

**Point 3:**
Coherence (figure 7) is a second-order statistics and it certainly changes a lot. (Please don't show all the irrelevant scatter in the plots, just the smoothed coherences.)

You are right, the scatter should not be shown here. Fig. 7 was modified accordingly.

I don't think this large change in coherence towards a much more pointed shape has been observed anywhere in atmospheric measurements while the theoretical von Karman coherences have been verified for small separations at several occasions. Please comment on this. Since the two-point cross spectra changes so drastically you should also expect the auto-spectra to change (figure 6).

Figure 7 shows the coherence in $\kappa_1$-direction of the three velocity components separated by a spatial distance $\triangle x_2$ in $x_2$-direction in comparison to the Mann model. The Mann model is based on the von Kármán energy spectrum, which is reflected by the coherence of the Mann-modelled wind field. For coherence in $x_2$-direction with respect to spatial steps in $x_3$-direction, the coherences of

the Mann model and the Time-mapped Mann model fit well, as shown in the appendix in figure A.2. However, for the coherence in $x_1$-direction, ,you are right, the spectrum shows significant differences to the Mann model. That shows, the time-mapping goes at the cost of the spatial correlations in $x_1$-direction which is expected since the $x_1$-axis is stretched and compressed which is also reflected by the spectrum in $x_1$-direction in figure 6(a). Lines 384-385 and lines 388-390 were changed.

This is obscured in figure 6 in the way the spectra are treated and plotted. First of all, it is customary, and with good reason, to plot the pre-multiplied spectrum $(\kappa_1 F_1(\kappa_1))$ because it makes it easier to see how the variance is distributed on frequency. Secondly, please do bin averaging so you plot an equal number of power spectral densities per decade. In this way it is possible to see the differences between the conventional and time-mapped spectra.

Thank you for the comment. Fig. 6 is re-poltted with $(\kappa_1 F_1(\kappa_1))$ on the y-axis and the noise in the plot was removed as advised.

Also regarding figure 6, I think it is totally unphysical to assume uncorrelated time mapping at every point and it obviously give nonsensical $F_2(\kappa_2)$ and $F_3(\kappa_3)$ spectrum. There are no reason to show these.

Concerning the uncorrelated Time-mapped Mann field, there seems to be a misunderstanding. It is not uncorrelated time-mapping in each point of the slice. The slices are mixed up so that the spatial correlation in the plane is destroyed, but still all points in the $x_2$-$x_3$-plane are shifted by the same time step in the temporal direction. This explained in lines 336-337 of the marked version of the manuscript.
But you are right, that at least in the new versions of figure 6 (b) and (c) plotting the uncorrelated version is not reasonable anymore, so we removed it from there.

**Point 4:**
There seems to be no good explanation on the very non-intermittent behavior of blade root flapwise moment in figure 10. It is also hard to understand the behavior of the kurtosis in figure 11. Why do you see very regular peaks at $\tau$ equal to interger seconds?

Thank you for this remark. The plot in figure 10a does not reveal intermittency which is actually there but with another frequency than for the other loads. This goes back to the frequency of the loads. The energy spectra of the thrust, torque, and tower fore-aft bending moment show peaks at the 3P frequencies and higher whereas the blade root flapwise bending moment also shows a peak for the 1P frequency. This carries over to the kurtosis spectrum shown in figure 11, which shows also the 3P frequency for the thrust. For the blade root flapwise bending moment, the frequency of fluctuations relates to the 1P frequency of the load energy spectrum. A non-monotonically decreasing kurtosis is also visible in Schwarz's dissertation (figure 5.8), which confirms our result that the intermittency is not strictly descreasing. However, the results in Schwarz dissertation are not completely comparable because the CTRW time series in there are either fully correlated or not correlated at all. The dependence of the oscillations in the kurtosis plots on the intensity of the spatial correlations in the $x_2$-$x_3$-plane is subject to a follow-up paper that we are currently working on. That means there is intermittency, it was not visible with the specific choice of the time step.

In order to clarify this more in the manuscript, we have added plots for the energy spectra of the blade root flapwise moment and the thrust in the appendix. Further, we have adapted the choice

of the time step for the plotting in figure 10a in order to show the intermittency more clearly. To clarify this, we have modified the manuscript in lines 441-447

**Specific remarks**

**1.)** In the introduction, which is nice, it would be great to be more specific on what the different studies show. Please state how large are the difference in percent instead of stating "very close", "agree quite well", "are different from" wherever it is possible.

Thank you for the remarks. The mentioned sentences are now modified to give a better description of the results of the reviewed researches in the introduction section. The lines 46-47 and 55-57 in the marked version of the manuscript were modified accordingly.

**2.)** Is there any physical reason for the choice the distribution of time increment maybe related to the fact that it is $\alpha$-stable? Eq (17) seems to miss "p()" on the left hand side.

Thank you for the notification. Yes, equation (17) indeed describes a probability density function (pdf), $p(\tau_\alpha)$.

An important reason for the special choice Fig.10(a) of the distribution is that for the time shift in the original CTRW model, it has to be ensured that the time step does not become negative. This means according to Kleinhans & Friedrich (Continuous-time random walks: Simulation of continuous trajectories, Phys. Rev. E 76, 2007) that the distribution for the temporal increments has to be a "fully skewed stable distribution".
The pdf for $\tau_\alpha$ in our equation (17), which we adapted from Kleinhans & Friedrich, is given in general form in (Metzler & Klafter: The random walk's guide to anomalous diffusion: A fractional dynamics approach, Physics Reports 339, 2000) in their equation (C.11) in *proposition C.3* (adapted to our notation):

$$p_{\alpha,\beta}(\tau_\alpha) = \frac{1}{\pi} \text{Re} \int_0^\infty dz \exp\left(-i\tau_\alpha z - z^\alpha \exp\left(i\frac{\pi\beta}{2}\right)\right). \tag{1}$$

Kleinhans & Friedrich use this equation for the case $0 < \alpha \leq 1$ and $\beta = -\alpha$ (for $\beta \neq 0$, this distribution is skewed).
Metzler & Klafter state in their *proposition C.11* that $p_{\alpha,-\alpha}(\tau_\alpha) = 0$, for all $\tau_\alpha < 0$. This ensures that there are no negative time steps. Further, the asymptotic behavior of this distribution for large $\tau_\alpha$ is determined by $\alpha$, $p_{\alpha,-\alpha}(\tau_\alpha) \sim \tau_\alpha^{-(1+\alpha)}$ (proposition C.5).

In order to enable the reader of our manuscript to look deeper into the derivation of the used form of the pdf of $\tau_\alpha$, we have also added the source for the general derivation of the pdfs to the manuscript in lines 221-222 and line 228.

**3.)** Smaller language issues. l 37 "realistic" → "realistically", l 110, I think it is more correct to use "component" instead of "direction", l 112 "statistics are" → "statistics is" (also l 319), l 115 "independent from" → "independent of", l 127 "coh" should be "coh", l 225 A distribution is not delta correlated but it can have a delta destribution.

Thank you for the comments, all language issues are now corrected. Regarding the delta-correlated fields, since the idea of the completely uncorrelated field follows the work of Schwarz

et al., we have followed the exact wording they have used in their work. However, we have changed this confusing wording to "uncorrelated field" in the manuscript.

**4.)** Eq (13) A square is missing on the $\kappa$ after the Kronecker delta symbol.
Thank you. Equation (13) is now corrected according to your comment.

---

## Author Response (AR3)

**Author's Responses 1**

Thank you for the valuable feedback on our manuscript. We addressed the remarks in the comments below.

Please note that the reviewer's comments are written in green, and the authors' responses are written in black.

Please also note that all line numbers in the responses refer to the line numbers in the **marked version** of the manuscript.

This is a comment on my previous review and how I think the authors have responded:

**1.:** It is great to see the large change in figure 1. It really shows that the intermittence is small scale, not general for all scales. This has really improved the manuscript. The second point for this comment is also well addressed.

Thank you for your positive feedback. Your comment has really helped us to improve this figure and to correct the equations.

**2.:** The authors argue why they limit their investigations to isotropic turbulence. This certainly limits the scope of the paper but it is acceptable.

We agree with you, it is certainly a limitation of the scope of the paper. However, we wanted to show how the time-mapping affects the statistics and highlight the effects on the loads.

**3.:** The way the spectra are shown is definitely improved. But it gives rise to another question: Why are your theoretical spectra offset from the simulated spectra? it is around 50% in the inertial sub-range and similar in the energy-containing range except at the very lowest $\kappa_1$. This has to be explained.

Thanks for pointing out this issue. We have checked our process carefully and repeated it for a wide range of model input parameters to ensure the correctness of our workflow. First of all, in figures 6-b and 6-c, the x-axis was $\kappa_2$ and $\kappa_3$ respectively. These figures have been corrected and plotted against $\kappa_1$ instead.

That brings up the second issue which is the considerable difference in the inertial sub-range. The reason behind such a shift is the turbulence field scaling process needed to achieve the target $TI$. On the other hand, the Mann-box field without scaling fits the theoretical curve as expected (see attached Fig. 1 below). Fig. 6 in the paper has been corrected and only the non-scaled field is shown which matches the theoretical curve. Also the same was done in Fig. A1 in the appendix.

[Figure]

Figure 1: Comparison of averaged spectra in (a) $x_1$-, (b) $x_2$-, and (c) $x_3$-directions for the Mann field ($F_i^{\text{Mann}}(\kappa_i)$), a rescaled Mann field e.g.: ($F_i^{\text{rescale}}(\kappa_i)$) and the theoretical spectra ($F_i^{\text{theory}}(\kappa_i)$)

One last point here is the difference at the low $\kappa$ values $< 10^{-3}$. That can be attributed to the smoothing which estimates the average based on an averaging window. Since there are a limited number of points in this range, the average isn't representative. We tried different average windows (see the attached Fig.2 below) and included in the paper what we think is a good compromise between the clarity of the figure and the accuracy.

[Figure]

Figure 2: Comparison of spectra in $x_1$ direction with smoothing using different averaging windows = (a) 1 point (no averaging), (b) 10 points, and (c) 50 points

All the points discussed above are limited to Fig. 6 and they are mainly visualization issues of the turbulent field. That does not influence the used fields in the analysis and does not affect the results of the paper. The smoothing was done to improve the visual comparison between the theoretical model and the Mann-turbulent field. These modifications and further explanation of the new Fig. 6 are shown in lines 331-380 in the marked version of the paper.

**4.:** This part is also improved.

Thank you for your positive feedback

**5.:** I think the conclusion reflect the changes well.

Thank you for your positive feedback. Your comments have really helped us to improve the conclusion to be more clear.

**6.:** All in all, the authors have done significant improvements to the manuscript. It is a step in the way of obtaining non-Gaussian inflow fields, but there is still a long way to go. I think the

manuscript is worthy of publication provided that the authors solve the issue with the offset between the simulated and theoretical, isotropic spectra.

Thank you for your positive feedback. We hope that the improvements in Fig. 6 and in the corresponding paragraphs in the paper have answered your questions. Thank you for your patience and effort.

Best regards, Jakob Mann

**Author's Responses 2**

Thank you for the valuable feedback on our manuscript. We addressed the remarks in the comments below.

Please note that the reviewer's comments are written in green, and the authors' responses are written in black.

Please also note that all line numbers in the responses refer to the line numbers in the **marked version** of the manuscript.

I would like to thank the authors for the answers to my remarks and questions to the revised version of the manuscript, the addition of Appendix B and the modification of Fig. 10, in which the time lags considered have now been selected in such a way that heavy tails are also visible for the RootFlap moments!.
The authors have completed Table 3 with the information on the turbine model and it is now clear that a tilt angle and gravity effects have been considered in the model, each of which produces deterministic 1P load variations for each blade. However, this supplementary information and the updated result in Fig. 10 raises a new fundamental question for me. The adaptation of the considered time lags in Fig. 10a has led to a sudden appearance of heavy tails in the root-flap moments that were not visible in the results of the first two versions of the manuscript. In my opinion this means that the heavy tails now visible in Fig. 10a are obviously caused by the deterministic 1P loads and not by the intermittency of the wind field. Fig. B1 in Appendix confirms the prominent 1P peak as well as the higher harmonics in the spectrum of the root-flap moments. A further indication that the heavy tails visible in Fig. 10a may be caused by deterministic loads results from the fact that the heavy tails strongly decay for higher time lags, while they are still clearly visible for the wind field (Fig. 8). A comparable behavior of decaying heavy tails for larger time lags can found in Fig. 10 for the their loads, so that also here it can be suspected that the visible heavy tails could be caused by deterministic 3P loads and not by the wind field.
If the heavy tails visible in the results for the time-mapping Mann model are caused by the deterministic loads, the question arises why the calculations with the original Mann model do not show heavy tails at all for any time lag (Fig. 10). Actually, I do not understand these results and would like to ask the authors to carefully check the calculations and the setups and provide more explanations. Are the turbine models really consistent? For the time being and with the available information and results, I cannot agree tostro the main conclusion of the manuscript (line 479ff), according to which the intermittency of the wind field generated by means of the time-mapped Mann model is transferred to the rotor loads. In order to demonstrate the influence of the intermittency of the wind field and to justify this conclusion, I strongly suggest to first consider a rotor without tilt and without gravity forces, i.e. without deterministic load variations, for both wind fields and I would be happy if my open questions and concerns could be solved.

We agree on the fact that heavy tails shown in Fig. 10 are closely related to the deterministic P loads. However, as you also pointed out, the heavy tails in Fig. 10 for the four loads, as well as the values of the kurtosis in Fig. 11 show a clear difference between the Mann and the time-mapping Mann models. Therefore, even though the intermittent behavior of the loads is related to the P frequencies, it is not purely caused by gravitational forces.
We have verified the comparison between the Mann and the time-mapping Mann to ensure the correctness of our calculations' correctness and identical set-ups for the numerical simulations. Furthermore, we have obtained similar results when comparing Mann and the time-mapping Mann wind

fields for other operational conditions, i.e. rotational speed, size of the turbine, mean wind speed, and turbulence intensity. In all cases, only the time-mapping Mann wind field induces heavy tails on the PDFs, and values of kurtosis are higher than 3 when calculating the increments of the loads. Such behavior is not observed with any of the Mann wind fields for the different simulations. Thus, we claim that the intermittency on the loads derives from the interaction of both, the intermittency of the wind field and the dynamics of the turbine.